# Direct on-swab metabolic profiling of vaginal microbiome host interactions during pregnancy and preterm birth

Pamela Pruski[1,15], Gonçalo D. S. Correia[1,2,15], Holly V. Lewis[3,4,5], Katia Capuccini[3,4], Paolo Inglese [1,2], Denise Chan[3,4,5], Richard G. Brown[4,5], Lindsay Kindinger[4,6], Yun S. Lee[3,4], Ann Smith [7], Julian Marchesi[1,3], Julie A. K. McDonald [8], Simon Cameron [1,9], Kate Alexander-Hardiman[1], Anna L. David[6], Sarah J. Stock [10], Jane E. Norman[10,11], Vasso Terzidou [3,4,12], T. G. Teoh[13], Lynne Sykes [3,4,5], Phillip R. Bennett [3,4,5,14], Zoltan Takats [1,2,3✉] & David A. MacIntyre [3,4,14✉]

The pregnancy vaginal microbiome contributes to risk of preterm birth, the primary cause of death in children under 5 years of age. Here we describe direct on-swab metabolic profiling by Desorption Electrospray Ionization Mass Spectrometry (DESI-MS) for sample preparation-free characterisation of the cervicovaginal metabolome in two independent pregnancy cohorts (VMET, $n = 160$; 455 swabs; VMET II, $n = 205$; 573 swabs). By integrating meta-taxonomics and immune profiling data from matched samples, we show that specific metabolome signatures can be used to robustly predict simultaneously both the composition of the vaginal microbiome and host inflammatory status. In these patients, vaginal microbiota instability and innate immune activation, as predicted using DESI-MS, associated with pre-term birth, including in women receiving cervical cerclage for preterm birth prevention. These findings highlight direct on-swab metabolic profiling by DESI-MS as an innovative approach for preterm birth risk stratification through rapid assessment of vaginal microbiota-host dynamics.

[1] Division of Systems Medicine, Department of Metabolism, Digestion and Reproduction, Faculty of Medicine Imperial College London, London, UK. [2] National Phenome Centre, Imperial College London, London, UK. [3] March of Dimes Prematurity Research Centre at Imperial College London, London, UK. [4] Imperial College Parturition Research Group, Institute of Reproductive and Developmental Biology, Department of Metabolism, Digestion and Reproduction, Imperial College London, London, UK. [5] Queen Charlotte's & Chelsea Hospital, Imperial College London, London, UK. [6] Elizabeth Garrett Anderson Institute for Women's Health, University College London, London, UK. [7] Faculty of Health and Applied Sciences, University West of England, Bristol, UK. [8] MRC Centre for Molecular Bacteriology and Infection, Imperial College London, London, UK. [9] School of Biological Sciences, Institute for Global Food Security, Queen's University Belfast, Belfast, UK. [10] MRC Centre for Reproductive Health, University of Edinburgh, Edinburgh, UK. [11] Faculty of Health Sciences, University of Bristol, Bristol, UK. [12] Chelsea & Westminster Hospital, NHS Trust, London, UK. [13] St Mary's Hospital, Imperial College Healthcare NHS Trust, London, UK. [14] Tommy's National Centre for Miscarriage Research, Imperial College London, London, UK. [15] These authors contributed equally: Pamela Pruski, Gonçalo D. S. Correia. ✉email: z.takats@imperial.ac.uk; d.macintyre@imperial.ac.uk

The vaginal microbiome is a key mediator of reproductive tract pathophysiology. Unlike other mucosal surfaces such as the gut, low diversity in the vaginal microbiome and dominance by *Lactobacillus* species is considered a hallmark of health, particularly during reproductive years. *Lactobacillus* species depletion and increased microbial diversity are characteristic of bacterial vaginosis (BV)[1], and associates with both the increased risk of acquisition[2] and ineffective treatment of sexually transmitted infections, including HIV[3,4]. High diversity, BV-associated vaginal microbiota have also been linked to HPV infection and cervical dysplasia[5,6], in vitro fertilization failure[7], miscarriage[8] and preterm birth[9–13]. Current clinical diagnosis of vaginal infection is largely limited to subjective assessment of clinical symptoms in addition to time-consuming microscopic evaluation of vaginal swab samples, pH and, in some cases, culture[14–16]. These approaches fail to detect many clinically relevant species and lack insight into overall community composition. Such information can be obtained using next-generation sequencing-based approaches (e.g. metagenomics and metataxonomics), but these involve complex and time-consuming sample preparation, data processing and expense that prevents their use for routine bedside testing. Further, molecular-based characterization of microbiota is unable to assess microbiota:host interactions that ultimately determine health and disease phenotypes. Therefore, there exists a need for rapid, point-of-care vaginal diagnostics to facilitate faster clinical decision making, more judicious use of antibiotics and targeted treatment strategies.

A common mechanistic pathway linking sub-optimal vaginal microbiota composition (VMC) and pathophysiology is activation of host-innate immune response and inflammation[17], which can be suppressed by *Lactobacillus* species such as *L. crispatus*, through the modification of the metabolic *milieu* of the cervicovaginal mucosa[16,18]. During pregnancy, untimely activation of inflammation in gestational tissues (e.g. cervix, foetal membranes) caused by ascending vaginal infection is thought to cause a significant proportion of infection-associated preterm birth[19,20], which remains the primary cause of death in children under 5 years of age[21]. Consistent with this, we have recently reported that cervical cerclage, a procedure used to reinforce the cervix in women at risk of preterm delivery due to cervical shortening, can induce vaginal bacterial dysbiosis, inflammatory activation and premature cervical ripening associated with increased risk of preterm birth, if performed using a braided instead of monofilament cerclage material[22]. In contrast, *L. crispatus* dominance of the vaginal niche during pregnancy has been associated with protection against preterm birth[10,12,22]. Despite this, individual and ethnic variation in VMC and host response[11,23,24] as well as cost and time constraints have limited the utility of current analytical methods used for vaginal microbiota characterization to inform clinical decision-making during pregnancy. Current methods for preterm birth prediction have poor positive predictive value and fail to provide insight into underlying aetiology, which may explain low efficacy of interventions designed to prevent preterm birth[25].

We have recently described a method that enables rapid, objective assessment of the chemical composition of mucosal surfaces using sample preparation-free, direct on-swab desorption electrospray ionization mass spectrometry (DESI-MS)[26,27]. This involves directing a pneumatically assisted electrospray of charged aqueous droplets directly onto a rotating swab, where it forms a liquid film that desorbs and ionizes molecules from the sample, which are transferred to a mass spectrometer via an atmospheric pressure ion-transfer line. With this method, metabolic profiles can be acquired rapidly (in <3 min) and directly from a swab sample without the need for laborious sample preparation or chemical extraction procedures. While lack of chromatographic separation limits assay selectivity, it provides the advantage of vastly simplifying the amount of maintenance and operator intervention required. Additionally, since the solvent stream only ablates sample from a small area of the swab, the method is virtually non-destructible supporting multi-assay use from the sampling device (e.g., cytokine/immune marker profiling or 16S rRNA sequencing)[16]. Collectively, these characteristics make direct on-swab DESI-MS particularly suitable for deployment in clinical point-of-care settings.

We hypothesized that vaginal microbiota:host interactions would be reflected in the metabolic milieu of the cervicovaginal mucosa during pregnancy and provide sufficient biochemical information to predict both bacterial composition and host immune response. To test this, direct on-swab DESI-MS profiling was used to characterize the cervicovaginal metabolome of two independent cohorts of women prospectively sampled throughout pregnancy (VMET, $n = 160$; 455 swabs; VMET II, $n = 205$; 573 swabs, Fig. 1A). These data were then integrated with matched metataxonomic and immuno-profiling data to identify DESI-MS metabolic signatures predictive of VMC and local inflammatory status. We then examined if these predictive models could be used to monitor changing VMC and host inflammatory responses that associate with preterm birth and clinical interventions (e.g., cervical cerclage) used during pregnancy.

## Results

**Baseline characteristics of the prospective study subjects**. A total of 160 women recruited to the VMET study were longitudinally sampled throughout pregnancy ($n = 455$ swabs) (Fig. 1A). These samples were all subjected to metabolic profiling by DESI-MS and a cross-platform comparison using five complementary liquid chromatography-MS (LC-MS) assays (Fig. 1A). A second independent patient cohort ($n = 205$, VMET2 study) were longitudinally sampled at comparable time points ($n = 573$ swabs). Samples from these patients were analysed by DESI-MS and a subset were used for immuno-profiling studies. Key demographics including maternal age, BMI, gravida and ethnicity were similar between the two patient cohorts (Table 1). Preterm births rates in these high-risk populations were 18% ($n = 29$; VMET) and 21% ($n = 44$; VMET2).

**Prediction of VMC by direct on-swab DESI-MS**. Ward's linkage hierarchical clustering (HC) analysis with Jensen–Shannon distance metric of bacterial species (operational taxonomic counts, OTUs) was used to group vaginal samples into 11 distinct groups as evaluated with Silhouette scores (Supplementary Fig. 1) that were subsequently reduced into community state types (CSTs) consistent with previous studies (Fig. 1B)[9,12,13,22,23,28]. Six major CSTs were identified across both cohorts: (1) CST I (*L. crispatus*); (2) CST II (*L. gasseri*); (3) CST III (*L. iners*); (4) CST IV (*Gardnerella vaginalis*); (5) CST V (*L. jensenii*); (6) CST VII (other *Lactobacillus* spp.). CST VI (*Bifidobacterium breve*) was identified in the VMET2 and not in the VMET cohort, due to differences in the primer sets used for amplicon generation (Supplementary Fig. 1)[29,30]. Higher taxonomic classification at genera level was achieved by grouping samples from CST I, II, III, V, and VII into *Lactobacillus*-dominated (LDOM, VMET: $n = 379$; VMET2: $n = 427$), and samples from CST IV and VI into *Lactobacillus*-depleted (LDEPL, VMET: $n = 49$; VMET2: $n = 112$) categories (Supplementary Table 1).

Using linear mixed effect modelling, a total of 113 metabolite features determined in DESI-MS negative and positive ion modes were found to significantly discriminate between LDOM and LDEPL in independent analyses of both patient cohorts (Fig. 1B)

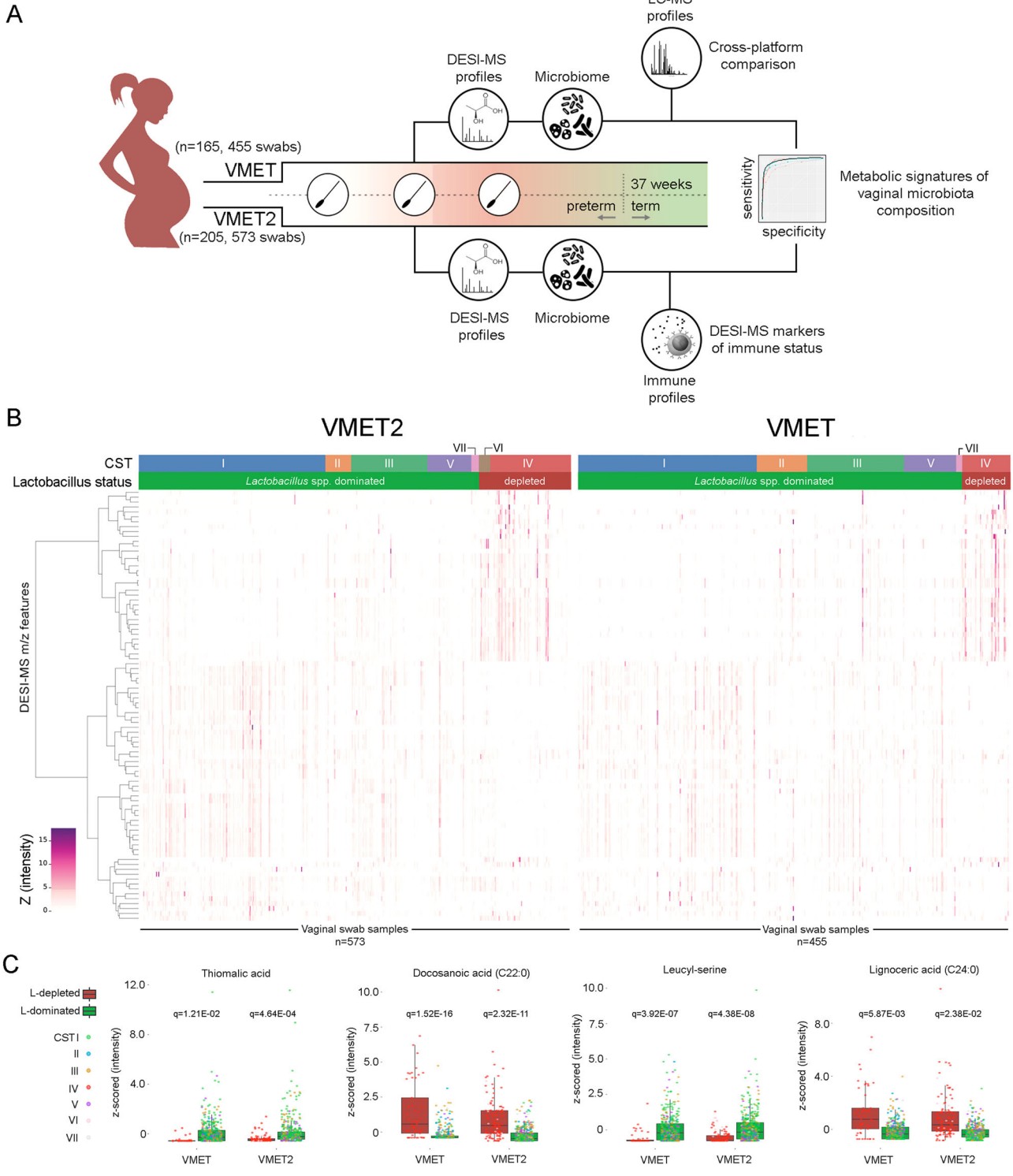

**Fig. 1 Direct on-swab desorption electrospray ionization mass spectrometry (DESI-MS) metabolic profiling of cervicovaginal fluid enables robust prediction of vaginal microbiome compositions. A** Study design and longitudinal multi-omic sampling and analysis workflow of cervicovaginal swab samples collected from two independent pregnancy cohorts (VMET: $n = 165$, 455 swabs; VMET2: $n = 205$, 573 swabs). Data from each cohort were analysed independently, and features selected only if their Benjamini–Hochberg $q$-value was smaller than 0.05 in both datasets, after matching the hits from each analysis by their $m/z$ values. **B** Heatmaps representing relative concentrations of DESI-MS (negative mode)-derived metabolic features ($n = 88$) significantly differing between *Lactobacillus* spp. dominated (L-dominated, green) and *Lactobacillus* spp. depleted (L-depleted, red) states in both independent patient cohorts (see Table S2). **C** Boxplots of representative discriminatory metabolic features with corresponding Benjamini–Hochberg $q$-values identified including thiomalic acid, leucyl-serine, docosanoic acid (C22:0), lignoceric acid (C24:0) with calculated $z$-score measured in the two patient cohorts VMET2 (left, $n = 203$, 539 swabs) and VMET (right, $n = 160$, 428 swabs). The lower and upper bounds of the box represent the 25th and 75th percentile values, respectively, and the interior horizontal line the median value. Whiskers are drawn from the corresponding box boundary to the most extreme data point located within the box bound ± 1.5 × IQR (interquartile range). $m/z$ mass-to-charge ratio, *CST* community state type.

**Table 1 Demographics and clinical characteristics of the study cohorts.**

| Cohorts | VMET | | | | VMET2 | | | | |
|---|---|---|---|---|---|---|---|---|---|
| Demographics | sPTB | Term | Total | P value | iPTB | sPTB | Term | Total | P value |
| Subject, n | 29 (18%) | 131 (82%) | 160 | | 3 (1%) | 41 (20%) | 161 (79%) | 205 | |
| Sample, n | 79 | 376 | 455 | | 9 | 105 | 459 | 573 | |
| Mean maternal age, years ± SD (range) | 32.2 ± 6.4 (20–43) | 22.7 ± 7.0 (8.7–36.7) | 22.5 ± 7.0 (8.7–36.7) | p = 0.17 | 38.0 ± 3.1 (34.0–41.0) | 32.7 ± 5.9 (20.0–44.0) | 33.2 ± 4.7 (20–43) | 33.3 ± 4.9 (20–44) | p = 0.50 |
| Mean sample gestation, weeks (±SD range) | 21.8 ± 6.8 (11.7–34.3) | 33.0 ± 5.0 (19–49) | 32.9 ± 5.2 (19–49) | p = 0.15 | 23.4 ± 7.2 (15.3–34.5) | 21.5 ± 6.4 (12.2–32.6) | 22.4 ± 6.8 (11.0–35.6) | 22.3 ± 6.8 (10.5–36) | p = 0.12 |
| Mean BMI, kg/m² (±SD, range) | 26.2 ± 4.4 (18–36) | 24.2 ± 4.4 (18–48) | 24.5 ± 4.5 (18–48) | p = 0.0002 | 28.9 ± 3.5 (24.0–32.0) | 25.5 ± 5.6 (20.0–46.1) | 25.3 ± 5.1 (18.1–39.9) | 25.3 ± 5.1 (18.1–46.1) | p = 0.19 |
| Ethnicity, n/N (%) | | | | p = 0.14 | | | | | p = 0.66 |
| Caucasian | 17/113 (15%) | 96/113 (85%) | 113/160 (71%) | | 2/130 (2%) | 20/130 (15%) | 108/130 (73%) | 130/205 (63%) | |
| Asian | 1/11 (9%) | 10/11 (91%) | 11/160 (7%) | | 0/28 (0%) | 8/28 (29%) | 20/28 (71%) | 28/205 (14%) | |
| Black | 11/35 (31%) | 24/35 (69%) | 35/160 (22%) | | 1/35 (3%) | 10/35 (29%) | 24/35 (69%) | 35/205 (17%) | |
| Others | 0/0 (0%) | 0/0 (0%) | 0/160 (0%) | | 0/12 (0%) | 3/12 (25%) | 9/12 (75%) | 12/205 (6%) | |
| Not recorded* | 0/1 (0%) | 1/1 (100%) | 1/160 (0%) | | 0/0 (0%) | 0/0 (0%) | 0/0 (0%) | 0/205 (0%) | |
| Gravida | | | | p = 0.01 | | | | | p = 0.21 |
| 0 | 2/50 (4%) | 48/50 (96%) | 50/160 (31%) | | 0/1 (0%) | 1/1 (100%) | 0/1 (0%) | 1/205 (0%) | |
| 1 | 8/34 (24%) | 26/34 (76%) | 34/160 (21%) | | 0/45 (0%) | 5/45 (11%) | 40/45 (89%) | 45/205 (22%) | |
| 2 | 5/26 (19%) | 21/26 (81%) | 26/160 (17%) | | 1/59 (2%) | 15/59 (25%) | 43/59 (73%) | 59/205 (29%) | |
| >3 | 14/50 (28%) | 36/50 (72%) | 50/160 (31%) | | 2/100 (2%) | 20/100 (20%) | 78/100 (78%) | 100/205 (49%) | |
| Previous mid trimester loss/PTB | | | | p = 0.00004 | | | | | p = 0.001 |
| No | 7/93 (8%) | 86/93 (92%) | 93/160 (58%) | | 1/86 (1%) | 7/86 (8%) | 78/86 (91%) | 86/205 (42%) | |
| Yes | 22/67 (33%) | 45/67 (67%) | 67/160 (42%) | | 2/119 (2%) | 34/119 (29%) | 83/119 (70%) | 119/205 (58%) | |
| Previous excisional cervical treatment | | | | p = 0.045 | | | | | p = 0.029 |
| No | 24/147 (16%) | 123/147 (84%) | 147/160 (92%) | | 0/105 (0%) | 15/105 (14%) | 90/105 (86%) | 105/205 (51%) | |
| Yes | 5/13 (38%) | 8/13 (62%) | 13/160 (8%) | | 1/100 (1%) | 28/100 (28%) | 71/100 (71%) | 100/205 (49%) | |
| Cervical cerclage material intervention | | | | p = 0.22 | | | | | p = 0.0007 |
| Braided | 10/45 (22%) | 35/45 (78%) | 45/160 (28%) | | 1/26 (4%) | 13/26 (50%) | 12/26 (46%) | 26/205 (13%) | |
| Monofilament | 7/23 (30%) | 16/23 (70%) | 23/160 (14%) | | 0/30 (0%) | 8/30 (27%) | 22/30 (73%) | 30/205 (15%) | |
| Abdominal | 0/0 (0%) | 0/0 (0%) | 0/160 (0%) | | 0/2 (0%) | 0/2 (0%) | 2/2 (100%) | 2/205 (1%) | |
| No intervention | 12/92 (13%) | 80/92 (87%) | 92/160 (58%) | | 2/142 (1%) | 18/142 (13%) | 122/142 (86%) | 142/205 (69%) | |
| Not recorded* | 0/0 (0%) | 0/0 (0%) | 0/160 (0%) | | 0/5 (0%) | 2/5 (40%) | 3/5 (60%) | 5/205 (2%) | |

sPTB spontaneous preterm birth, iPTB iatrogenic preterm birth. Comparisons of means between sPTB and Term group were done using two-sided Student's t-tests for continuous covariates (maternal age, sample gestation and BMI), and two-sided Pearson's chi-squared tests were used to compare the distribution of categorical variables (ethnicity, gravida, previous mid trimester loss/PTB, previous excisional cervical treatment, cervical cerclage material intervention).

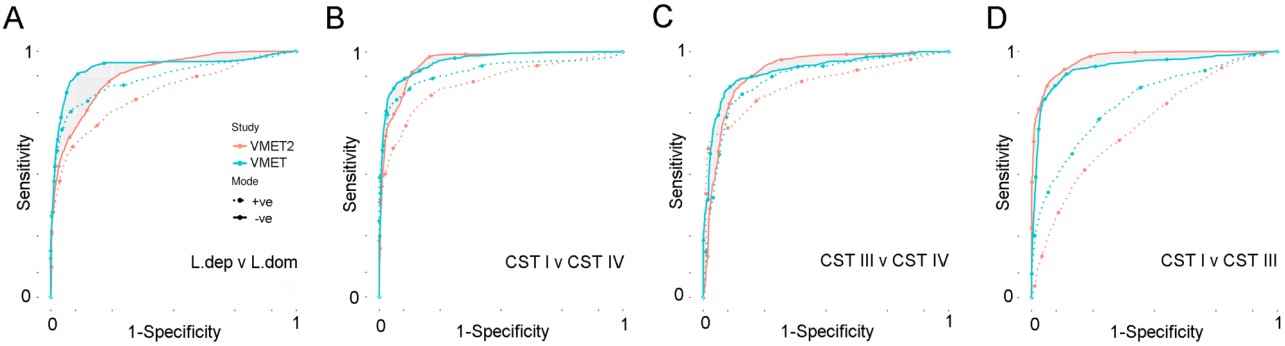

**Fig. 2 Comparison of DESI-MS classification performance between different vaginal microbiome compositions. A** ROC-curve analysis showing performance of direct swab analysis by DESI-MS operating in both negative and positive ion polarity modes to predict *Lactobacillus*-depleted vaginal microbiome compositions in both the VMET (blue; AUC: 94.1, sensitivity: 62.0, specificity: 97.8) and VMET2 (red; AUC: 90.6, sensitivity: 54.5, specificity: 96.4) patient cohorts. Discrimination between the major vaginal community state types (CST) could also be readily achieved using DESI-MS across both patient cohorts, including **B** CST I vs IV, **C** CST III vs IV and **D** CST I vs III. Overall, predictive performance of DESI-MS was comparable to that of models constructed from LC-MS assays (Supplementary Table 3 and Supplementary Fig. 4).

(Supplementary Data 1). The vast majority of these metabolites did not change with advancing gestation (Supplementary Fig. 2). A detailed assessment of variance explained by CST, gestational age, ethnicity, BMI and maternal age for each of the metabolite biomarkers highlighted CST and ethnicity as being the major factors of variance (Supplementary Fig. 3). In addition, the proportion of total metabolome variance explained by each factor was estimated with permutational multivariate analysis of variance (PERMANOVA, Supplementary Table 2). While the $R^2$ values estimated for each covariate varied between metabolic profiling assays, their relative importance was consistent, with between-individual variability explaining the most variance (35–45.2%), followed by CST (2.2–25.2%), ethnicity (0.5–2.4%) and gestational age (0.4–1.8%). Of those, thiomalic acid and leucyl-serine were consistently higher in LDOM samples, whereas docosanoic acid and lignoceric acid were significantly higher in LDEPL samples (Fig. 1C). Random forest classifier and ROC-curve analysis were then used to assess the performance of DESI-MS to predict VMC (Fig. 2 and Supplementary Fig. 4). Robust prediction of genera-level classification (LDOM vs LDEPL) was observed across patient cohorts particularly using features derived from negative ion polarity mode (VMET/VMET2; AUC 94.1/90.6; sensitivity: 62.0/54.5; specificity: 97.8/96.4). Discrimination between the major vaginal CSTs (CSTI, III and IV) could also be readily achieved by direct on-swab DESI-MS (Fig. 2). The predictive performance of DESI-MS for all models was comparable to averaged (min/max) prediction performance of LC-MS assays (VMET/VMET2; AUC 95.76 (94.3–97.7); sensitivity: 69.74 (57.5–80.2); specificity 98.28 (97.7–98.9)) (Supplementary Table 3 and Supplementary Figs. 4 and 5).

**In vitro DESI-MS profiling of vaginal commensal and pathogenic bacterial isolates.** We next investigated if any of the discriminatory metabolites identified in the in vivo DESI-MS analyses could also be observed in culture isolates (n = 25) of bacterial species recognized as being predominant members of major vaginal CSTs. In total, 27 of the discriminatory metabolites identified in vivo were detected by DESI-MS in swabs of culture biomasses following correction for background media concentrations (Fig. 3). Approximately half of these metabolites were detected at levels lower than that observed in media background controls, whereas the remainder were found at levels higher than media background. No clear relationship was observed between the DESI-MS detected levels of these metabolites in vitro, CST membership nor levels observed in vivo.

**Assessment of host immune response at the mucosal interface using direct on-swab DESI-MS.** A panel of 22 soluble immune markers including cytokines, chemokines, immunoglobulins and members of the complement system were measured in a subset of the VMET2 cohort samples (n = 391). Random forest regression analysis was used to predict the log-transformed concentrations of each marker using DESI-MS-derived features. Robust prediction (cross-validated $R^2 > 0.25$) was observed particularly for IL-1β (CV $R^2 = 0.51$), IL-8 (CV $R^2 = 0.37$), C3b/iC3b (CV $R^2 = 0.31$), IgG3 (CV $R^2 = 0.31$), IgG2 (CV $R^2 = 0.27$) and MBL (mannose-binding lectin) (CV $R^2 = 0.26$) (Fig. 4A). Immune markers with CV $R^2 > 0.1$ (n = 9) and DESI-MS features with an $R^2 > 0.1$ for the linear regression against the immune marker (n = 23) revealed a metabolic signature strongly associated with local immune phenotype primarily characterized by altered levels of long-chain fatty acids, glycerophospholipids and ceramides (Fig. 4B and Supplementary Table 4). Both DESI-MS predicted and immunoassay-measured levels of C3b, IL-1β, IgG2 and IgG3 were elevated in *Lactobacillus*-depleted vaginal microbiomes indicating activation of the local innate and adaptive immune response (Fig. 4C).

**Direct-on swab DESI-MS as a potential tool for PTB risk stratification.** We next examined the relationship between pregnancy outcome, VMC and inflammatory status as predicted by DESI-MS by combining all available data from both VMET and VMET2 patient cohorts. High vaginal microbiome diversity and instability during pregnancy, as defined by shifts between *Lactobacillus*-dominated and *Lactobacillus*-depleted compositions classified with 16S rRNA gene-based metataxonomics, was associated with an increased risk of preterm birth compared to those women maintaining *Lactobacillus*-dominance throughout pregnancy (odds ratio (OR) 1.97, 95% confidence interval (CI) 1.03–1.66, p = 0.04) (Fig. 5A). Similarly, vaginal microbiota instability determined solely by DESI-MS was also associated with a higher risk of preterm birth although this did not reach statistical significance (OR 1.47, 95% CI 0.75–2.78, p = 0.25).

Increased levels of IL-1β, measured directly by immunoassay or predicted by DESI-MS, were associated with high-diversity VMC in both those women experiencing term birth or preterm birth. However, levels were significantly higher in preterm birth cases (preterm 59.63 pg/ml vs term 1.94 pg/ml, p = 0.006, Welch two sample t-test, Fig. 5B). Increased levels of MBL, again either measured directly by immunoassay or predicted by DESI-MS, were also observed in high-diversity VMC of women

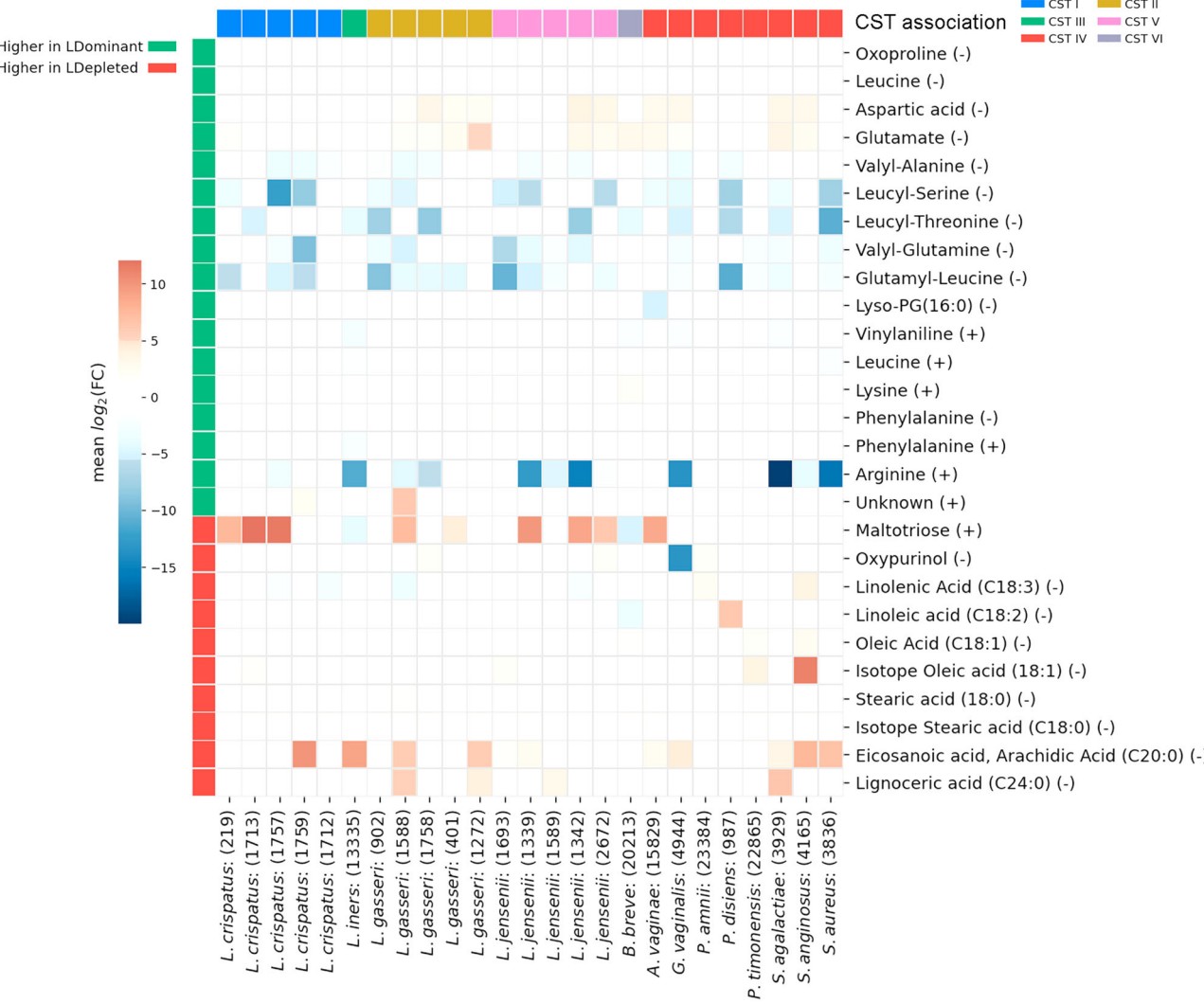

**Fig. 3 Detection of in vivo discriminatory metabolite features in bacterial biomasses by DESI-MS.** Discriminatory metabolites identified in the in vivo DESI-MS analyses were detected by DESI-MS in bacterial isolates ($n = 25$) of species recognized as being predominant members of major vaginal community state types (CST). A total of 27 metabolites were detected at levels lower or higher than that observed in media background controls, where the mean $\log_2$ fold change (FC) was estimated as the ratio of the mean intensity in the bacterial biomass samples to the mean intensity in the background culture media samples.

subsequently delivering preterm (term 0.042 ng/ml vs preterm 0.692 ng/ml, $p = 0.007$, Welch two-sample $t$-test, Fig. 5C).

We next tested if DESI-MS metabolic, metataxonomics and inflammatory marker profiles obtained at three different stages of gestation, could predict subsequent preterm birth. Overall, the predictive capacity of these models was poor (Supplementary Table 5). We therefore focused our subsequent analyses on preterm birth phenotypes more likely to associate with dysregulated vaginal microbiota–host interactions, as previously described[22]. In women at high risk of preterm birth due to cervical shortening, vaginal levels of MBL were most frequently increased in those treated with a cervical cerclage using braided suture material compared to those where cerclage had been performed with monofilament suture (10/11, 91% vs 9/21, 43%, $p = 0.011$, Fisher's exact test). This was similarly detected when VMC and MBL levels were estimated using only DESI-MS analysis of vaginal swabs (7/11, 64% vs 8/21, 38%, $p = 0.316$; Fig. 5D). Immunoassay also indicated increased IL-1β after braided cerclage insertion, but this was less consistently detected by DESI-MS prediction (Fig. 5E). However, following treatment with braided cerclage material, DESI-MS did accurately predict

low levels of IL-1β in women delivering at term compared to those who subsequently delivered preterm (Fig. 5F).

## Discussion

Despite recent developments in metataxonomics and metagenomics, VMC characterization in clinical settings remains largely limited to culture and microscopy, which like molecular-based approaches, fails to capture information regarding host response. Our method, which is easily amenable to bedside point-of-care testing[31,32], addresses this limitation by leveraging information contained within the cervicovaginal mucosa to provide robust detection of VMCs and simultaneous estimation of host immune and inflammatory status.

Many of the discriminatory metabolites identified in our study have been previously reported using gas chromatography-MS or LC-MS based assays in women suffering from BV[33–36] or in HPV infection[37] and have biologically plausible roles in mediating vaginal health and disease. For example, increased vaginal levels of short- and long-chain fatty acids and biogenic amines are associated with activation of pro-inflammatory pathways[36,38–41], which contribute to reduced barrier integrity of the epithelia and

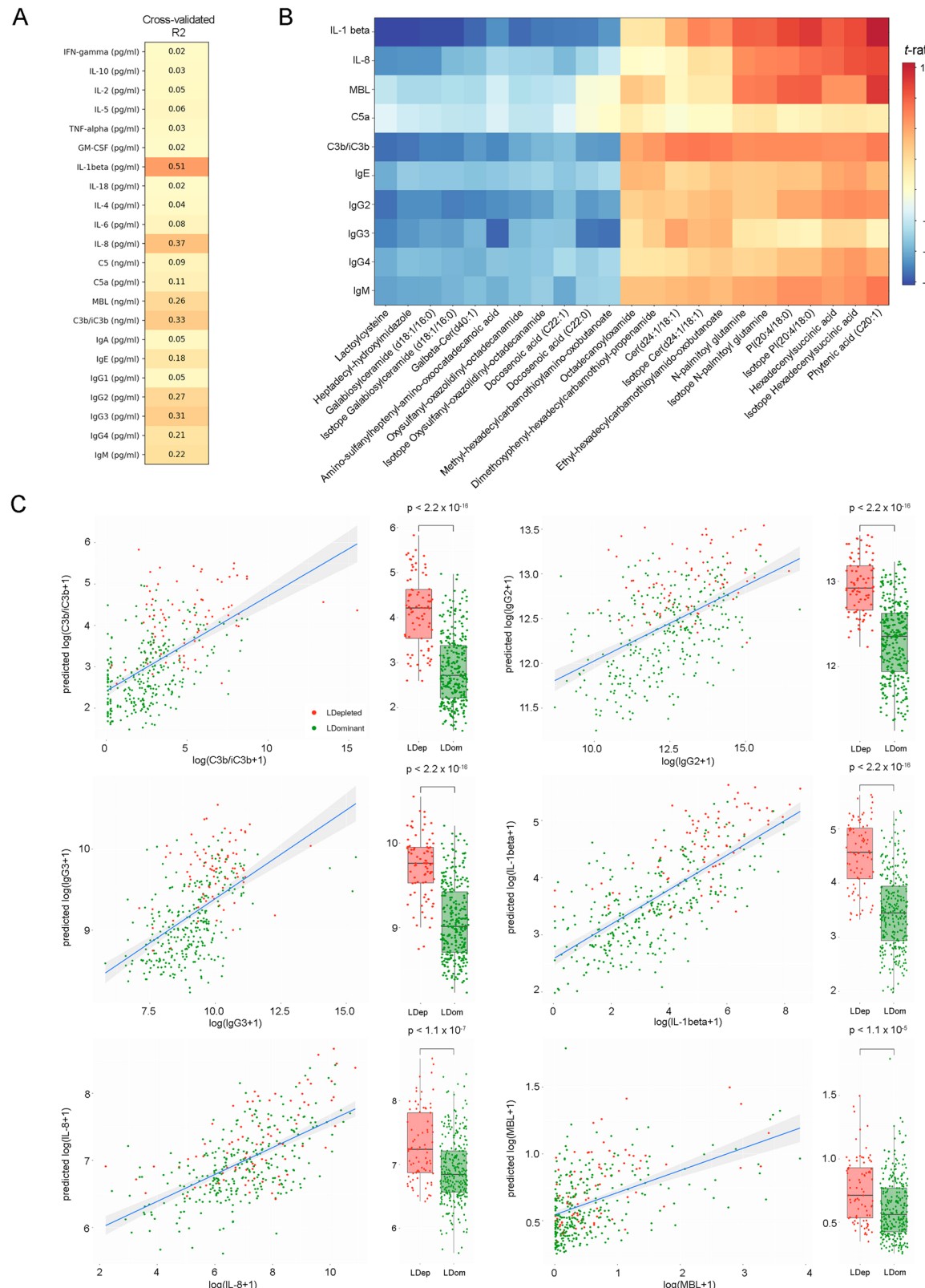

consequently, increased risk of infection[34,42–45]. Furthermore, cervicovaginal lipid species have recently been associated with tumour progression and genital inflammation defined using an aggregated score, in a study of 78 non-pregnant, HPV-negative/positive women with cervical dysplasia[37]. Lipid changes in vaginal discharge have also been associated with vulvovaginal candidiasis[46]. In our study, we profiled a larger panel of soluble immune mediators, including humoral response mediators, and identified associations with the metabolome via regression models. Despite different experimental approaches, our results provide further evidence that lipid species are a core component of the metabolic signature of inflammation and immune response in the vaginal niche. Some of the discriminatory markers used in our in vivo DESI-MS prediction models were also detected by DESI-

**Fig. 4 Assessment of host response at the mucosal interface using direct on-swab DESI-MS profiling. A** Cross-validated $R^2$ value for all 22 corresponding measured immune mediator concentrations. **B** Heatmap of top 23 significantly correlated metabolite features with top 10 immune mediators (IL-1β, IL-8, MBL, C5, C3b/C3bi, IgE, IgG2, IgG3, IgG4, IgM). *t*-ratio ranges from +10 (red) to −7.5 (blue). **C** Association between predicted log-transformed value of immune marker by DESI-MS and measured log-transformed values by multiplexed immune-assay for IL-1β (CV $R^2 = 0.51$), IL-8 (CV $R^2 = 0.37$), C3b/iC3b (CV $R^2 = 0.33$), IgG3 ($R^2 = 0.31$), IgG2 (CV $R^2 = 0.27$), MBL (CV $R^2 = 0.26$). A linear regression line was fitted to the log-transformed values and their corresponding prediction. A box plot of predicted immune marker levels for LDEP (red) and LDOM (green) samples is also presented ($n = 136$ pregnancies, 369 swabs). The lower, interior horizontal line, and upper bounds of the box represent the 25th, median and 75th percentile values, respectively. Whiskers are drawn from the corresponding box boundary to the most extreme data point located within the box bound ± 1.5 × IQR (interquartile range). *P* values are reported for a two-tailed Welch *t*-test for the difference in mean predicted immune markers between LDEP and LDOM.

MS in vitro swab analyses of bacterial isolate biomasses. It should be noted that the bacterial isolates analysed here were not representative of the genetic diversity of vaginal commensal and pathogenic strains; however, our analyses indicate that the metabolite signatures predictive of microbiota composition and inflammation in vivo are likely derived from both bacterial and host sources.

Some studies have estimated that around 23–30% of total cervicovaginal metabolic variation is associated with bacterial composition[36,47,48]. Similar estimates obtained in our data emphasize the impact of the microbial composition on the metabolome, which is greater than the effect of gestational age, ethnicity, maternal age, or BMI. However, using the repeated measures per individual, we found that the top proportion of variance explained could be attributed to between-individual variability. The magnitude of the $R^2$ values for microbial composition was higher in the LC-MS assays compared to DESI-MS. This may represent greater coverage of relevant constituents of the metabolome that are not detected by DESI-MS or may reflect the technical variability of DESI-MS which, like other ambient ionization methods, is recognized to be larger than that of LC-MS where implementation of experimental quality control (QC), instrumental drift correction and data filtering procedures are more established[49]. Despite this, our findings highlight the capacity of direct on-swab DESI-MS to rapidly capture this information from the metabolome without the need for laborious sample preparation and comparatively high per/assay time and economic costs associated with coupled chromatography-MS assays for similar applications. This lends itself to customization and automation for ease-of-use, which are important characteristics for point-of-care testing[32]. Our approach also offers the advantage of providing objective simultaneous assessment of microbiota composition and inflammatory state. In comparison, current 'gold standard' diagnostic methods for vaginal infection (e.g. Amsel criteria) are limited to a combined subjective assessment of clinical symptoms and microscopy grading of bacterial morphotypes as well as selective culture.

There is now substantial data supporting a role for the vaginal microbiome and host immune responses in shaping preterm birth in a proportion of women[10–12,22]. However, prediction of preterm birth using DESI-MS metabolic profiles, metataxonomics or inflammatory marker data in our patient cohort was poor. This is not surprising given the fact that preterm birth is a multi-aetiological disease state that can be caused by many different factors, including non-microbial and non-immune related causes[19,20]. Because of this, we focused subsequent analyses on women who receive cervical cerclage with braided suture, who we have previously shown are at increased risk of preterm birth that involves a phenotype characterized by vaginal dysbiosis and local immune activation[22]. Here we confirmed these findings in an independent patient population and highlight the utility of direct swab analysis by DESI-MS to rapidly detect both microbiota and inflammatory changes caused by the intervention that associate

with subsequent preterm birth risk. The ability to provide such information at point-of-care would be transformative for directing clinical decision making and ultimately improving outcomes for these women and their babies.

In this study, DESI-MS prediction of VMCs was capable of monitoring vaginal microbiome diversity and instability long-itudinally throughout pregnancy, which in the study participants was associated with increased risk of preterm birth. These results are consistent with a recent meta-analysis of metataxonomics-based studies of the vaginal microbiome in pregnancy, which reported higher variance of VMC across trimesters in women subsequently experiencing preterm birth[50]. An additional strength of our approach is the ability to simultaneously capture relationships between VMC and local immune response. Innate immune activation in the vagina often accompanies disease states associated with suboptimal VMC, including BV, preterm birth, and sexually transmitted infections[51–53]. For example, *L. crispatus* dominance of the vaginal niche is associated with suppressed levels of IL-1β compared to high-diversity compositions[17,54–58]. Further, MBL is a recognition molecule for *G. vaginalis*[59] and several single-nucleotide polymorphisms in the *MBL2* gene have been reported to increase the risk of BV[60,61]. Our findings show that local vaginal immune responses are reflected in the cervicovaginal metabolic phenotype and can be readily detected by DESI-MS. This has clear translational benefit. We have recently reported that cervical cerclage, a procedure used to reinforce the cervix in women at risk of preterm delivery due to cervical shortening, can induce vaginal bacterial dysbiosis, inflammatory activation and premature cervical ripening associated with increased risk of preterm birth, if performed using a braided instead of monofilament cerclage material[22]. Consistent with these findings, DESI-MS showed capacity to detect clinically relevant inflammatory responses to cerclage insertion that associated with patient outcome. Coupled with VMC prediction, this highlights direct swab profiling by DESI-MS as an innovative platform for host–microbiota interactions during pregnancy that could potentially facilitate PTB risk stratification, optimization of preventative interventions[29] (e.g. targeted antibiotic treatment following preterm premature rupture of the foetal membranes and response to treatment interventions during pregnancy[9]). While our study has focused upon application of DESI-MS in pregnancy cohorts, the rapid detection of VMC and host response offered by this methodology could also inform treatment strategies in other clinical scenarios. For example, efficacy of the topical pre-exposure prophylactic, Tenofovir, is superior in preventing HIV acquisition in women with *Lactobacillus*-dominated VMCs compared to those dominated by *G. vaginalis* and BV-associated bacteria[3]. It is therefore conceivable that efficacy and effectiveness could improve through stratification of treatment using point-of-care metabolic profiling of vaginal swabs. By extension, our approach may also be useful for bedside monitoring of response to treatments designed to optimize VMC, such as live biotherapeutics[62] and vaginal microbiome transplantation[63].

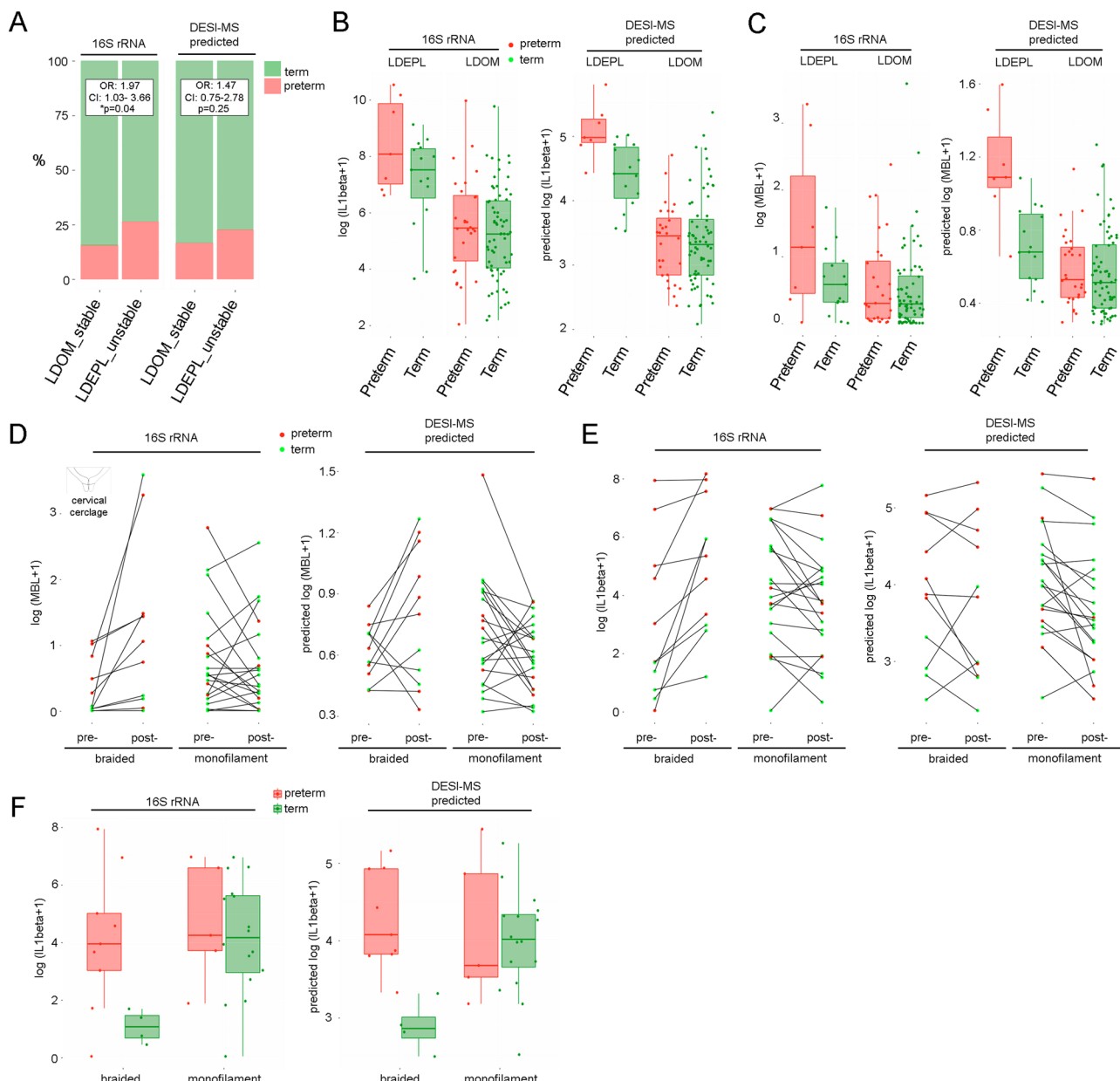

**Fig. 5 Vaginal microbiome instability and immune activation associates with preterm birth risk and poor outcomes following cervical cerclage.**
**A** Increased risk of PTB (red) was associated with vaginal microbiome instability (defined by shifts between *Lactobacillus* spp.-dominated (LDOM) and *Lactobacillus* spp.-depleted (LDEPL) compositions) measured by 16S rRNA-based metataxonomics (OR 1.97, 95% CI 1.03–3.66, $p = 0.04$, two-sided mid-$p$ exact test) or predicted using DESI-MS profiles (OR 1.47, 95% CI: 0.75–2.78, $p = 0.25$, two-sided mid-$p$ exact test). **B** LDEPL vaginal composition was associated with increased IL-1β levels compared to LDOM; however, highest levels were observed in LDEPL women subsequently having preterm delivery. This relationship was also observed when IL-1β levels and vaginal microbiota composition were predicted using direct swab profiling by DESI-MS ($n = 103$ pregnancies, 103 swabs). **C** A relationship between LDEPL, increased MBL and subsequent preterm birth was also detected by DESI-MS profiling ($n = 103$ pregnancies, 103 swabs). **D** Elevated MBL and **E** elevated IL-1β levels were observed in response to cervical cerclage performed with braided cerclage material, but not monofilament material ($n = 34$ pregnancies, 68 swabs). **F** Preterm birth in women treated with cervical cerclage using braided cerclage material was associated with higher IL-1β levels compared to term birth outcomes ($n = 13$ pregnancies, 13 swabs), whereas no relationship between IL-1β levels measured or DESI-MS-predicted were observed with pregnancy outcome following cervical cerclage using monofilament material ($n = 21$ pregnancies, 21 swabs). All box and whisker plots are drawn with the lower, horizontal interior line, and upper bounds of the box representing the 25th percentile, median and 75th percentile values, and whiskers extending from the lower or upper box bonds to the position of the most extreme data point within ± 1.5 × IQR (interquartile range).

A limitation of our approach was the relatively low sensitivity of prediction. Further inspection of the 16S rRNA gene amplicon data indicated that this was largely associated with misclassification of samples harbouring mixed compositional structures, often containing *G. vaginalis* (Supplementary Fig. 6). This

suggests that hard-clustering techniques often used for determining CSTs may under-estimate the impact of low abundance taxa on the host mucosal metabolome. Comparison of predictive performances from the LC-MS assays indicated that metabolic coverage may also impact misclassification rate with small polar

or non-polar molecule-based assays marginally outperforming lipid-based assays (Supplementary Figs. 5 and 7). It is also worth noting that the analysis of VMC in this study was limited to the assessment of relative abundances of the bacterial component of microbiome and a comparatively small number of patients carrying less prevalent community compositions (e.g., *L. gasseri*, *L. jensenii* and *B. breve*-dominated). The detection of *B. breve*-dominated women in our study was limited to the VMET2 patient cohort where a mixed formulation of the 27F forward primer set was used. This primer formulation has been shown to maintain the rRNA gene ratio of key vaginal species including *Lactobacillus* spp. to *Gardnerella* spp. as well as improve detection of *Bifidobacterial* species, which are otherwise not detected by the 'universal' 27f primer often used in metataxonomics studies[30,64]. Consistent with these results, a number of recent studies have also identified *B. breve*-dominated vaginal microbiota using meta-taxonomics approaches[65,66]. Additional datasets that provide improved representation of more diverse compositions and rarer taxa may also eventually facilitate prediction of VMC by DESI-MS beyond major CSTs as described here, and toward species level classification. It is also important to note that recent meta-taxonomics studies have indicated higher false discovery rates when using relative abundance from NGS data compared to quantitative PCR[67]. Future studies incorporating microbial load within a sample may further improve the sensitivity of DESI-MS predictive models, particularly those with intermediate compositions.

In conclusion, we show that direct on-swab DESI-MS permits rapid and robust assessment of vaginal microbiota:host immune interactions reflected within the cervicovaginal mucosal metabolome. While we show that this may offer an approach for preterm birth risk stratification and selective targeting of preventative treatments, we expect it to have wider application to the assessment of vaginal microbiota–host interactions in other clinical scenarios, including both pregnant and non-pregnant women.

## Methods

**Study subjects and sample collection.** The study was conducted with approval of the NHS National Research Ethics Service (NRES) Committees London-City and East (REC 12/LO/2003) and London–Stanmore (REC 14/LO/0328), and by the North of Scotland Research Ethics Service (REC 14/NS/1078). All participants provided written informed consent prior to sampling and experiments were performed in accordance with the approved institutional guidelines. Recruitment and sampling were performed at Imperial College Healthcare NHS Trust Hospitals (Queen Charlotte's and Chelsea and St Mary's Hospitals), London, UK, at Chelsea & Westminster Hospital (NHS Trust, London, UK), University College London Hospital (NHS Foundation Trust, London, UK) and the Royal Infirmary of Edinburgh, Scotland, UK. Eligibility criteria were pregnant women with a singleton pregnancy, with and without risk factors for preterm birth. Exclusion criteria included women under 18 years of age, sexual intercourse within 72 h of sampling, vaginal bleeding in the preceding week, antibiotic use in the preceding 2 weeks, multiple pregnancies, HIV or hepatitis C-positive status. Detailed maternal clinical metadata were collected for all participants and birth outcome recorded following delivery. Preterm birth was defined as labour prior to 37 weeks of gestation. Cervicovaginal fluid swab samples were collected at up to five timepoints throughout pregnancy (1. 8–16, 2. 16–20, 3. 20–26, 4. 26–30, 5. 30–37 weeks) from the posterior fornix using BBL CultureSwab MaxV Liquid Amies swabs (Becton, Dickinson and Company, Oxford, UK), placed in either Amies transport media or a sterile microcentrifuge tube on ice, before long-term storage at −80 °C.

**Metabolic profiling of cervicovaginal swabs using direct swab analysis by DESI-MS.** All chemicals used were analytical reagent grade. HPLC grade methanol and water for DESI-MS analysis and swab sample extraction were purchased from Sigma-Aldrich (St Louis, MO). The method used for the direct analysis of vaginal swab samples by DESI MS is detailed elsewhere[26]. Briefly, we used an LTQ-Orbitrap Discovery mass spectrometer (Thermo Scientific, Bremen, Germany) coupled with a DESI-MS source designed for direct swab analysis. Swabs were placed into a rotating holder positioned orthogonally in front of the MS inlet capillary with a swab–capillary distance of approximately 2 mm. The DESI sprayer tip was pointed to the swab centre with a tip–sample distance of 1.5–2 mm and a distance between the tip and the inlet capillary of 2 mm. The entire surface of the medical swabs was analysed by DESI-MS through clockwise rotation of the swab

toward the MS capillary. The cervicovaginal mucosa was absorbed from the swab tip by gently desorbing the analytes with charged droplets of methanol/water (95:5, v/v) mixture and directed to the mass spectrometer. For each sample, 30 scan mass spectra ($m/z$ 50–1000, $R = 30,000$ (FWHM)) were recorded in the negative and positive ion mode.

**Metabolic profiling of cervicovaginal swabs using direct swab analysis by LC-MS analysis.** Liquid extraction was performed on each swab by adding a MeOH:H$_2$O (1:1, v-v) solution as eluent to a final concentration of 50 mg vaginal fluid/ml. Each blank swab was extracted with 1 ml solution using a repeated sonication and vortexing cycle for 30 s each. Recovery of soluble material was achieved through centrifugation of swabs ($2000 \times g$ for 2 min) seated in a 200 µl loading tip positioned in a sterile microcentrifuge tube. Associated supernatants were pooled and centrifuged at $16,000 \times g$ for 10 min to remove insoluble material before the resulting supernatant was divided into three aliquots. An additional extraction of lipids from the swab was performed by repeating the procedure with an isopropanol:water (4:1, v-v) solution to a final extraction of 25 mg vaginal fluid/ml. The resulting extract was pooled with one aliquot of the methanol:water extract. All extracts were evaporated using a SpeedVac for further reconstitution before analysis by LC-MS[68,69].

*Reversed phase LC-MS analysis of small metabolites.* Sample reconstitution was performed in 300 µl of water. In all, 250 µl aliquot of reconstituted sample material were used for individual sample preparation of 96-well plates including the addition of 25 µl of full RP 2× labelled standard mix (L-glutamine-$^{13}$C$_5$; L-glutamic acid $^{13}$C$_5$; creatinine-methyl-D$_3$; cytidine-5,6-D$_2$; citric acid $^{13}$C$_6$; L-isoleucine-$^{13}$C$_6$$^{15}$N; L-leucine-13$^{13}$C$_6$; L-phenylalanine-$^{13}$C$_9$$^{15}$N; hippuric acid-D$_5$, benzoic acid-$^{13}$C$_6$, octanoic acid-$^{13}$C$_8$, L-tryptophane-$^{13}$C$_{11}$$^{15}$N$_2$). In addition, 50 µl aliquots of each sample were used for pooling and generation of QC sample. For chromatographic separation a 2 µl aliquot of extracted metabolites from each sample was injected onto a reverse-phase 150 × 2.1 mm ACQUITY 1.8-µm High Strength Silica (HSS) column (Waters Corp.) kept at 45 °C using an ACQUITY UPLC system (Waters Corp.). The mobile phase consisting of 0.1% v/v formic acid (Fisher Scientific) in water (A) and acetonitrile containing 0.1% formic acid (B, Sigma-Aldrich). Each sample was resolved for 12.65 min at a flow rate of 0.5 ml/min. The gradient consisted of 99% A and 1% B for 0.1 min, a ramp of curve 6–100% B from 0.1 to 10.70 min.

*Reversed phase LC-MS analysis for lipids.* Sample reconstitution was performed in 300 µl of water/isopropanol (4:1). In all, 250 µl aliquot of reconstituted sample material were used for individual sample preparation of 96-well plates including the addition of full RP-labelled standard mix (C17:0; LPC(6:0/0:0); LPC(9:0/0:0); LPC(15:0/0:0); PC(11:0/11:0); PC(15:0/15:0); PE(15:0/15:0); PA(17:0/17:0); PG(15:0/15:0); PS(17:0/17:0); SM(d18:1/17:0); Cer(d18:1/17:0); DG(19:0/0:0/19:0); PC(23:0/23:0); TG(8:0/8:0/8:0); TG(10:0/10:0/10:0); TG(12:0/12:0/12:0); TG(14:0/14:0/14:0); TG(15:0/15:0/15:0); TG(16:0/16:0/16:0); TG(17:0/17:0/17:0); TG(18:0/20:4/0:0); DG(18:0/18:0)). In addition, 50 µl aliquots of each sample were used for pooling and generation of QC sample. For chromatographic separation a 2 µl aliquot of extracted metabolites from each sample was injected onto a Waters Acquity UPLC BEH C8, 1.7 µm, 2.1 × 100 mm column (Waters Corp.) kept at 55 °C using an ACQUITY UPLC system (Waters Corp.). The mobile phase consisting of water:isopropanol:acetonitrile (50:25:25, all high-grade LC-MS solvents from Fisher Scientific or Sigma-Aldrich) + 5 mM ammonium acetate + 0.05% acetic acid + 20 µM phosphoric acid (A) and isopropanol:acetonitrile 50:50 (Sigma-Aldrich) + 5 mM ammonium acetate + 0.05% acetic acid (B). Each sample was resolved for 13.15 min at a flow rate of 0.5 ml/min. Starting conditions were 99% A and 1% B and the gradient changed with a ramp of curve 6 as follows: decrease to 70% A and 30% B over the first 2 min, decrease to 10% A with 90% B from 2 to 11.50 min, decrease to 0.1% A with 99.9% B from 11.50 to 12.50 min, after which the solvent composition returned to starting conditions over 0.1 min until 13.15 min.

*HILIC LC-MS analysis.* Sample reconstitution was performed in 200 µl of water:-acetonitrile mixture (1:3.6, v-v). 115 µl aliquot of reconstituted sample material were used for individual sample preparation of 96-well plates including the addition of 2.5 µl of 48× labelled IS standard mix ((uracil-2-$^{13}$C$^{15}$N$_2$; N-benzoyl-d$_5$-glycine, adenosine-2-D$_1$, adenine-2-D$_1$, taurine-$^{15}$N, L-tryptophan-D$_5$, L-phenyla-lanine-$^{13}$C$_9$$^{15}$N, creatine-(methyl-D$_3$)-monohydrate, L-arginine-$^{13}$C$_6$-hydro-chloride). In addition, 20 µl aliquots of each sample were used for pooling and generation of QC sample. HILIC chromatography analysis was performed using a Waters Acquity UPLC BEH HILIC (1.7 µm, 2.1 × 150 mm) column (Waters Corporation, Milford, MA, USA) kept at 40 °C. The mobile phases consisted of 0.1% formic acid and 20 mM ammonium formate in water (A, Fisher Scientific), and 0.1% formic acid in acetonitrile (B, Sigma-Aldrich) at a flow rate of 0.5 ml/min. Starting conditions were 5% A and 95% B and the gradient changed with a ramp of curve 6 as follows: increase to 20% A and 80% B over the first 5.5 min; increase to 50% A with 50% B from 5.5 to 7.1 min, after which the solvent composition returned to starting conditions over 0.1 min until 12.65 min. The injection volume was 2 µl of extracted metabolites for negative ion mode analysis.

*LC-MS instrumental operation and analysis*. The column eluent was introduced directly into the mass spectrometer by electrospray. MS was performed on a Waters Xevo G2-S QTOF mass spectrometer (Waters Ltd, Elstree, UK) operating in either negative-ion (ESI−) or positive-ion (ESI+) electrospray ionization mode with a capillary voltage of 1 kV for RP and 1.5 kV for HILIC, and a sampling cone voltage of 20 V. The desolvation gas flow was set to 1000 l/h and the desolvation temperature was set to 600 °C. The cone gas flow was 150 l/h, and the source temperature was 120 °C. Accurate mass was maintained by introduction of LockSpray interface of Leucine Enkephalin ($m/z$ 556.2771 in ESI+, $m/z$ 554.2615 in ESI−) at a concentration of 200 pg/μl in 50% aqueous acetonitrile with a scan time of 0.07 s over 4 scans, after each interval of 60 s. Data were collected in centroid mode from 50 to 1200 $m/z$ in MS scanning mode. To ensure system suitability and stability, a study reference (SR) QC sample was prepared by combining equal aliquots of all the samples and injected at regular intervals throughout the analytical run. This SR sample was also used to condition the column (30 injections) prior to the analysis of both the ESI+ and ESI mode batches. Blank samples (i.e. injection of the reconstitution solvent) were also run to check the presence of artefact or contaminant peaks. A dilution series of the SR sample was also acquired at the beginning and end of injection sequence. Data-dependent acquisition (DDA) and MS$^E$ analysis of the SR sample were performed for structural elucidation.

**Mass spectral data processing**. DESI-MS raw mass spectral data were converted from.raw files into.mzXML format via the ProteoWizard msConverterGUI[70]. The first 30 spectra per sample were averaged into a single spectrum and saved as.csv file using the MALDIquant R package (v1.19.3). The detailed parameter settings and algorithm used in the R package MALDIquant[71] are as follows: Peak detection was performed using 'detectPeaks', with SNR set to 3 and half window size to 10, and using the median absolute deviation (MAD) method. Peaks were aligned with a half window size set to 20, tolerance set to 0.002, signal-to-noise-ratio (SNR) set to 2 using the warping method LOWESS. The reference peaks were created by calling the function 'referencePeaks' from MALDIquant with method 'strict', 'minFrequency' set to 0.01 and peak binning tolerance 0.002.

Raw LC-MS data files were first converted to the.mzML open format and the Proteowizard msconvert..mzML files were processed in R (v4.0.3) using the XCMS (v3.10.2) package[72]. Peaks were detected with the centWave algorithm (ppm = 25, mzdiff = 0.001, prefilter = c(4, 1000), noise = 100, snthresh = 5), and grouped with the 'density' method (bw = 3, mzwid = 0.0007, minFrac = 0.4). The centWave peakwidth parameter was set depending on the chromatographic method (c(1.5, 14) for HILIC-LC-MS (−) c(3, 12) in RP-LC-MS Lipid (+/−) and c(1.5, 5) in the RP-LC-MS Metabolite (+/−)). Non-detected peaks were filled with fillPeaks, using the default arguments, and no retention time correction was done. The XCMS datasets were further filtered and the feature intensities drift corrected in Python (v3.8.5) using the nPYc-Toolbox (v1.2.4)[73]. Drift correction was performed using LOWESS trend-line model fitted on the SR samples. Features where the coefficient of variation measured on repeated injections of the SR sample were larger than 30% or whose Pearson correlation with dilution was less than 0.7 (measured using the SR dilution series) were excluded from the data matrix.

**Metabolite identification**. Target $m/z$ features were putatively annotated using online databases including HMDB, Metlin, MMCD, KEGG and Lipidmaps with a 5 ppm tolerance for each compound. To account for potential imprecision in $m/z$ measurement due to peak binning and data processing artefacts, raw $m/z$ value were confirmed by inspection of the.raw data. Metabolite annotation was performed by searching the measured $m/z$ ratios against METLIN (http://metlin.scripps.edu), Lipidmaps (http://www.lipidmaps.org) and the HMDB (http://www.hmdb.ca)[74–76]. Further structural elucidation was performed using MS/MS experiments via collision-induced dissociation on the LTQ-Discovery MS instrument (Thermo Scientific), and with DDA of the precursor ion by the quadrupole of the Xevo G2 XS Q-TOF mass spectrometer (Waters corporation). For the annotation of metabolites, the MS/MS spectra were matched against spectral libraries from HMDB, NIST and METLIN that were compiled with either authentic standards or theoretical assignment. Identification of metabolites where MS/MS reference spectra were not available were annotated using chemical fragmentation rules. For the structural assignment of glycerophospholipid, fragments of the polar head group or the fatty acyl chains were investigated to confirm the annotation proposed by the databases and discriminate isomers. MSI levels of compound identification were further reported as suggested by the Metabolomics Standards Initiative (MSI)[77].

**DNA extraction and sequencing of 16S rRNA amplicons**. Extraction of bacterial DNA was performed as previously described[78]. For the VMET cohort, V1–V3 hypervariable regions of bacterial 16S rRNA genes were amplified using a forward and reverse fusion primer. The forward primer was constructed with the Illumina i5 adapter (5′-AATGATACGGCGACCACCGAGATCTACAC-3′), an 8-base pair (bp) bar code, a primer pad (forward, 5′-TATGGTAATT-3′), and the 28F primer (5′-GAGTTTGATCNTGGCTCAG-3′) (64). The reverse fusion primer consisted of the Illumina i7 adapter (5′-CAAGCAGAAGACGGCATACGAGAT-3′), an 8-bp bar code, a primer pad (reverse, 5′-AGTCAGTCAG-3′), and the 519R primer (5′-GTNTTACNGCGGCKGCTG-3′). For the VMET2 cohort, the V1–V2 hyper

variable regions were amplified with the forward primer set (28f-YM) consisting of a mixture of the following primers mixed at a 4:1:1:1 ratio: 28F-Borrelia GAGTTTGATCCTGGCTTAG; 28F-Chlorflex GAATTTGATCTTGGTTCAG; 28F-Bifido GGGTTCGATTCTGGCTCAG; 28F GAGTTTGATCNTGGCTCAG. The reverse primer consisted of 388R TGCTGCCTCCCGTAGGAGT[30]. Sequencing was performed at RTL Genomics (Lubbock, TX, USA) using an Illumina MiSeq platform (Illumina Inc.). Resulting sequence data were analysed using the MiSeq SOP Pipeline of the Mothur package[79]. The Silva bacterial database (www.arb-silva.de/) was used for sequence alignment and classification was performed using the RDP (Ribosomal Database Project) database reference sequence files[80]. Determination of operational taxonomic unit taxonomies (phylum to genus) and species-level taxonomies was performed with USEARCH with 16S rRNA gene sequences from the cultured representatives from the RDP database[81]. Species-level taxonomies were complemented using information from the STIRRUPS database[82].

Counts from all OTUs assigned to the same species were summed to generate matrices of total counts per species. Species with less than 50 total counts were excluded. Community state types were then assigned to each sample with HC, using Ward-linkage and the Jensen–Shannon distance. The number of clusters which maximized the mean silhouette score were selected. The relative abundance of each species in a cluster was inspected and the clusters matched to CST reported in previous publications, by comparing the relative abundances of the top five OTUs obtained for each cluster. Samples where the most abundant species was a *Lactobacillus* spp. other than *L. crispatus*, *L. iners*, *L. gasseri* or *L. jensenii*, where manually assigned to a separate cluster (CST VII). HC analyses and heatmaps were performed using Python (v3.8.5), 'scipy'[83] (v1.5.2), the 'matptlotlib'[84] (v3.3.2) and 'seaborn'[85] (v0.11.0) libraries.

**Immune/inflammatory profiling of cervicovaginal samples**. Swabs were thawed on ice and re-suspended in 350 ml of phosphate-buffered saline solution containing protease inhibitor (5 μL/ml; Sigma-Aldrich) before being centrifuged at $7000 \times g$ for 10 min to remove cellular debris. Customized multiplex assays (R&D Systems) were used together with a Bio-Plex 200 system (Bio-Rad Laboratories Ltd) to quantify levels of interleukin-1β (IL-1β), IL-2, IL-4, IL-5, IL-6, IL-8, IL-10, IL-18, interferon-γ, GM-CSF and tumour necrosis factor-α, with a 10-fold dilution of 1L-8 performed using Calibrator Diluent RD6-52 prior to assaying. The Human Complement Magnetic Bead panel 1 (Milliplex®) was used for measurement of C5, C5a and MBL, while panel 2 was used for measurement of C3b. Levels of immunglobulins IgA, IgG1, IgG2, IgG3, IgG4 and IgM were assessed using the ProcartaPlex Human Antibody Isotyping Panel 7plex assay (Thermo Fisher Scientific).

**DESI-MS profiling of bacterial isolates**. Bacterial isolates analysed in this study were derived from the DSMZ-German Collection of Microorganism and Cell Culture GmbH (DSM number, Supplementary Table 6) or clinical samples received by the Imperial College NHS Healthcare Trust Diagnostic Microbiology laboratory (NHS number, Supplementary Table 6) at Charing Cross Hospital, London, after the completion of standard identification workflows. Identification of isolates from clinical samples was performed using a MALDI Biotyper instrument (Bruker, UK)[86]. Isolates were stored on beads and in glycerol broth at −80 °C. The bacterial culture library consisted of *Atopobium vaginae*, *Bifidobacterium breve*, *Gardnerella vaginalis*, *Lactobacillus crispatus* (5× isolates), *Lactobacillus gasseri* (5× isolates), *Lactobacillus iners*, *Lactobacillus gasseri* (5× isolates), *Lactobacillus jensenii* (5× isolates), *Prevotella amni*, *Prevotella disiens*, *Prevotella timonensis*, *Staphylococcus aureus*, *Streptococcus agalactiae* and *Streptococcus anginosus*. Microorganisms were grown from either fresh cultures or beads with each isolate cultured 10 times using optimal culture conditions (Supplementary Table 5).

Bacterial isolates were sampled directly from solid agar plates using a medical rayon swab before being transferred into a sterile microcentrifuge tube and stored at −80 °C. Three 'blank' swab samples of each solid agar media were also collected. Direct swab analysis using DESI-MS was performed in both positive and negative ion mode. Spectra were processed and assembled into a single data matrix. For each isolate, the intensities per $m/z$ feature of biological replicates were compared against their corresponding culture media background samples using the Wilcoxon–Mann–Whitney test. The list of significant features was matched to the set of identified features from the analysis of human cervicovaginal swab samples. The mean fold change for matched peaks (tolerance < 5 ppm) was estimated as the ratio of the mean intensity within bacterial biomass samples to the mean intensity in the background culture media samples. The heatmap in Supplementary Fig 6 was generated in Python (v3.8.5), with the 'matptlotlib' (v3.3.2) and 'seaborn' (v0.11.0) libraries.

**Statistical analysis of metabolomic profiling data**. The linear mixed-effects modelling analyses were performed in R using the 'lme4'[87] package (v1.1.25). For all metabolic profile variables in each data matrix, a linear mixed-effects model with the following 'lme4' formula was fitted: Metabolite ~ GestationalAge + CST + MaternalAge + BMI + Ethnicity + (GestationalAge||SubjectID). This model structure accounts for the repeated measures by using a random intercept per pregnancy and a random slope per pregnancy. No random effect term was used

to model correlation between the random slopes and random intercepts. Gestational age, CST, age, BMI and ethnicity were modelled as fixed effects. All models were fitted using restricted maximum likelihood ('REML = TRUE' in lmer function call). The 'emmeans'[88] package (v1.5.2.1) was used to perform contrast coding and obtain effect size estimates and $P$ values for contrasts and trends of interest. For the comparisons between different CST's and LDOM vs LDEPL, the mean levels and contrasts obtained from the model were estimated assuming maternal age = 30 years, BMI = 23 and gestational age = 20 weeks, and averaged over all levels of ethnicity. For the gestational age trend estimates were further averaged across all CSTs. Detailed tables with effect size estimates and $P$ values for these analyses and contrasts are available via a GitHub repository (see 'Code availability'). The comparisons between *Lactobacillus* dominant vs depleted was encoded as a comparison between the grand mean of CST I, II III, V and VII levels vs the mean of CST IV. The $P$ values for each contrast tested were calculated using the Kenward–Roger approximation, implemented in the 'pbkrtest' package (v0.4.8.6)[89]. The $P$ values for all contrasts within a single metabolic signal were not corrected for multiple testing. Instead, for each of the main contrasts of interest (early gestation vs late gestation, CST I vs III, CST I vs IV, CST III vs IV and LDOM vs LDEPL), all $P$ values obtained across all metabolic variables in an assay were pooled and FDR corrected together as a signature, using Benjamini–Hochberg false discovery rate correction and selecting a 5% FDR cut-off. The heatmaps in Fig. 1B were created in Python (v3.8.5), 'matptlotlib' (v3.3.2) and 'seaborn' (v0.11.0). Only features which were statistically significant after false discovery rate correction and replicated in both VMET and VMET2 datasets were plotted. To account for differences in $m/z$ calibration between datasets, a feature was only considered to be replicated between datasets if it was possible to find a marker with an $m/z$ error of less than 5 part-per-million in the final FDR corrected signature from the other study. The linear mixed effect model, semi-partial $r^2$ and conditional $R^2$ measures were calculated using the R packages 'r2glmm' (v0.1.2) and 'MuMIn' (v1.43.17), respectively. PERMANOVA analyses[90] were performed with the 'vegan' package (v2.5.7), using the adonis2 function, with method = 'euclidean', permutations = 999, and by adding terms sequentially following their order in the model formula log(MetaboliteMatrix + 1) ~ CST + GestationalAge + Ethnicity + BMI + Age + SubjectID.

For the immune-metabolite association analysis, a series of linear models with the formula lm(log(ImmuneMarker + 1) ~ Metabolite) were fitted in R (v4.0.3) for all immune-marker cytokine pair combinations, and the corresponding $F$-test $P$ values calculated. Benjamini–Hochberg false discovery rate correction was used independently for each immune marker analysis, with a 5% FDR cut-off. The heatmaps in Fig. 2A, B were generated using Python (v3.8.5), 'matptlotlib' (v3.3.2) and 'seaborn' (v0.11.0), using the results of the linear model analysis and the random forest regression models (detailed in the next section).

**Prediction of CST, preterm birth and immune markers from the metabolomic data**. The CST type was predicted from the metabolomic profiles using random forest classifiers. For the *Lactobacillus* dominant vs depleted comparison, samples which were assigned to CST I, II, III, V or VII were assigned to the *Lactobacillus* dominant class, and samples with CST IV or VI label were assigned to the *Lactobacillus* depleted. Random forest classifiers were trained to predict the CST or *Lactobacillus* depletion status using the metabolic profile variables, without using any other clinical or demographic variables, including gestational age. An assessment of the impact on model performance of using gestational age as covariate was also performed (see the analyses under 'CST Typing by DESI-MS' in the GitHub repository).

Prediction of preterm birth was also performed using random forest classifiers. The centred-log-ratio transformation was applied to the 16S data matrices with the R package 'propr' (v4.2.6)[91]. The gestational period was divided in three gestational windows (1st timepoint: 0–14 weeks; 2nd timepoint: 14–24 weeks; 3rd timepoint 24–40 weeks) and a random forest classifier model fitted at each window using only the 16S data, immune markers, or DESI-MS profiles as predictors. Only one sample per patient (selected as the closest to the gestational age window mid-point) was used in each model. For the immune marker prediction, immune marker concentrations were initially log-transformed after addition of a constant offset log($x$ + 1). A random forest regressor was trained to predict a single log-transformed immune marker level. All random forest models were trained in R (v4.0.3), using the 'randomForest' (v4.6-14)[92] package in combination with the 'caret' package (v6.0-86)[93], for model fitting, performance metric calculation and cross-validation. RF classifiers were trained using constant default parameters: number of trees 'ntree' = 1000, and the number of variables 'mtry = number of metabolic variables/3'. For the random forest regression models, 'mtry = sqrt(number of metabolic variables)'. A repeated ($n$ = 15) stratified fivefold cross-validation procedure was used for all models. This approach ensures that the train and test sets contain a similar proportion of samples from each class.

The performance of the RF models was assessed by calculating the $R^2$ using only data left out from each CV round test sets (performed by default by 'caret'). For the classifiers, ROC curves, precision–recall curves, accuracy, sensitivity, specificity, positive-predictive value and negative predictive values were calculated and plotted with the R packages 'caret', 'pROC' (v1.16.2)[94], 'plotROC' (v2.2.1)[95], 'precrec' (v0.11.2)[96] and 'ggplot2' (v3.3.2)[97]. For the regression analysis, the $R^2$ was calculated with 'caret' and predictions plotted with 'ggplot2'.

**Reporting summary**. Further information on research design is available in the Nature Research Reporting Summary linked to this article.

## Data availability

The metabolic profiling data generated in this study have been deposited and made publicly available in the MetaboLights database under study identifier MTBLS717. The sequence data for the study are publicly available through the European Nucleotide Archive [https://www.ebi.ac.uk/ena] under accession numbers PRJEB 11895, 12577 and (https://www.ebi.ac.uk/ena/browser/view/PRJEB41427). Relevant clinical and patient metadata are publicly available in the GitHub repository at https://www.github.com/gscorreia89/desims-cst-analysis/, digital resource identifier: https://doi.org/10.5281/zenodo.5513501[98].

## Code availability

Key analysis code and processed datasets required to reproduce the statistical analyses presented in this study are deposited and publicly available in the GitHub repository at https://www.github.com/gscorreia89/desims-cst-analysis/, digital resource identifier: https://doi.org/10.5281/zenodo.5513501[98].

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

## Acknowledgements
We would like to thank all patients who have participated in this study and members of the Women's Health Research Centre who facilitated and coordinated study recruitment and sample collection. We also acknowledge Lili Herendi, Verena Horneffer van der Sluis and Maria Gomez for coordinating and running LC-MS based assays at the Clinical Phenotyping Center (CPC) at Imperial College London, UK, and the NIHR Imperial Biomedical Research Centre (BRC), which provided infrastructure support. This work was supported by the European Research Council (Grant No. 617896), the Imperial College NIHR Biomedical Research Centre (Grant No. 58434), the Medical Research Council (Grant No. MR/L009226/1), NIHR Clinical Lectureship Scheme, Genesis Research Trust and the March of Dimes European Preterm Birth Research Centre at Imperial College London. A.L.D. is supported by the NIHR University College London Hospitals Biomedical Research Centre. The views expressed in this publication are those of the authors and not necessarily those of the NHS, NIHR or Department of Health and Social Care.

## Author contributions
P.P., G.D.S.C., L.S., P.R.B., Z.T. and D.A.M. conceived and designed the study. Clinical sampling and coordination of metadata was performed by H.V.L., D.C., R.G.B., L.K., A.L.D., S.J.S., L.S., V.T. and P.R.B. Bacterial isolates were obtained from S.C., K.A-H., J.A.K.M.D. and J.M. Experiments were performed by P.P., H.V.L., K.C., D.C., R.B., L.K., Y.S.L., J.A.K.M.D. and K.H. Data processing and analysis was performed G.D.S.C., P.P., P.I., A.S., Z.T. and D.A.M. P.P., G.D.S.C. and D.A.M. prepared all figures and tables and wrote the first draft of the manuscript. All authors critically reviewed, read and approved the final manuscript.

## Competing interests
P.P., P.R.B., Z.T. and D.A.M. hold patents for the use of rapid evaporative ionization mass spectrometry and desorption electrospray ionization mass spectrometry analysis of swabs and biopsy samples (US10026599B2, EP3265817B1). P.P., G.C., P.R.B., Z.T. and D.A.M. have filed a provisional patent for the use of rapid evaporative ionization mass spectrometry and desorption electrospray ionization mass spectrometry analysis of swabs for prediction of vaginal microbiota composition and inflammatory status (GB2110293.4). The remaining authors declare no competing interests.
