## [Peer Review File · Nature Communications]

REVIEWER COMMENTS

Reviewer #1 (Remarks to the Author):

This study investigates how the vaginal microbiome contributes to risk of preterm birth, the primary cause of death in children under 5 years of age, using metabolic profiling by DESI-MS for characterization of the cervicovaginal metabolome. The authors describe how the metabolome signatures can be used to predict the composition of the vaginal microbiome and inflammatory status. The primary findings focus on using on-swab metabolic DESI-MS profiling for rapid preterm birth risk stratification, a very important area of study.

The study design is generally good with two separate patient cohorts seen as a major positive. Although it was surprising that the identified metabolites were not further validated/quantified using a separate LC/QqQ based method, essentially the gold stand for such analyses. Yet the use of LC/MS QTOF data is compelling.

While I usually approach these desorption/ionization studies with a fair degree of skepticism, I am impressed with the level of rigor that went into this effort. While I would have preferred to see the QqQ analyses as a level of validation, nonetheless this is a solid paper. Although how the immunological data was used is also a bit confounding and how all the data from DESI, LC/MS, lipid profiling, combined with the immunological data was not clear.

The figures are well done and the data is compelling yet their interconnectivity is a bit confusing.

Page 3 desorp should be desorb

Reviewer #2 (Remarks to the Author):

This manuscript proposes a novel method to assess the state of the pregnancy vaginal microbiome for the purposes of predicting risk of pre-term birth. Results suggest that Desorption Electrospray

Ionization Mass Spectrometry (DESI-MS) can predict differences in vaginal microbiome composition (namely depletion of *Lactobacillus* spp.), community state type and inflammation. Additionally, the authors describe how DESI-MS could provide a risk estimation for pre-term birth, as they show the metabolome reflects shifts on vaginal microbiome and inflammation that are associated with PTB. While the method shows promise for rapid assessment of pre-term birth risk, the greatest challenge is that there is no direct assessment of the method for predicting pre-term birth (which should be possible with the cohorts assessed). It is also unclear whether the metabolic profiling technique will be more advantageous (in terms of sensitivity and specificity) at predicting risk of pre-term birth than previously established relationships such as cervix length, microbial composition, and inflammation.

Major Points:

1. It is unclear why DESI-MS was not used directly to predict pre-term birth within the cohorts assessed. Instead, DESI-MS measurements were used to predict bacterial composition/cytokine production, which was then discussed as an important factor in pre-term birth. DESI-MS does seem like a useful tool for gaining biological insight into the relationships between the microbiome and the host immune response in pregnancy, though this does not seem to be the focus of the manuscript. What is the motivation of using DESI-MS to predict immune marker concentration, and then using the predicted immune marker concentration to associate with pre-term birth classification?
2. If purpose of this manuscript is to introduce a new technology for diagnostic, it could be helpful to include more text in the introduction and discussion sections regarding why this methodology is superior to previous techniques (if any) to predict PTB or whether the goal is a new method to understand multi-omics relationships between metabolome, vaginal microbiome composition and inflammation. If the latter, how is it better than traditional techniques to quantify metabolome (RP-LC-MS, HILIC-LC-MS)? Additionally, if it is being proposed as a new diagnostic tool, how do prediction capabilities feasibility compare to the use of vaginal microbiota composition or a panel of immune biomarkers (say IL-1beta and MBL) to predict risk of PTB?
3. Were repeated measures considered in the random forest models? Pregnancy term influences vaginal microbiota composition and metabolite changes may also be dependent on time point in pregnancy.
4. For the selection of a signature to predict LDOM/LDEPL, how was the signature validated? Was all the data from both cohorts used in the linear mixed effect model that suggested the 113 metabolite features? Or was the signature validated in only a specific cohort?

5. For the in vitro measurements, why were multiple strains of *Lactobacillus* spp. evaluated compared to single strains of bacteria associated with CST IV? This may be important since *G. vaginalis* is a highly variable species and *L. gasseri* and *L. jensenii* were not observed as frequently in the samples. *L. iners* strains can also be quite variable and were the next most common *Lactobacillus* sp. after *L. crispatus* in the samples evaluated.

6. The discussion (p. 9) states that this approach could be extended for bedside treatments designed to optimize VMC. It could be interesting to assess the use of DESI-MS compared to standard practices to diagnose VMC (typically for BV) such as Nugent Score, Amsel Score, or 16S qPCR in regards to accuracy, cost, speed of result and ease of use by clinician (in clinic) or by patient (home self-swab). More discussion in this direction would be valuable.

Minor Points and Suggestions:

1. Table 1: define "i" and "s" PTB
2. Figure 1c: add statistical significance bars or p-values to plots.
- 3 Figure 4b: change DESI-MS m/z features to names of compounds.
4. Fig S2 and Fig S3: text on the plots is illegible.
5. Github link is not public (<https://www.github.com/gscorreia89/desi-ms-cst-typing/>).
6. It is unclear whether results support that DESI-MS allows for species level classification since the most resolution demonstrated was to predict CSTs, and thus only applies to *Lactobacillus* spp.

Reviewer #3 (Remarks to the Author):

The paper describes the use of a Direct on-swab metabolic profiling approach as a potential 'bedside' point of care testing technique for examining cervicovaginal microbiota and host immune activity related to pregnancy and preterm birth risk. It is an interesting and, potentially valuable addition to the literature. The manuscript is mostly well written and I only have several minor comments that should be addressed.

Line 50: "...increased >microbial< diversity..."

Lines 51-52 Not sure this is worded how you intend it to ("...and the effective treatment of STI...") - appears to suggest lactobacillus depletion/BV is associated with more effective treatment of STI

Lines 76 - 79: The influence of bacterial composition on the metabolome has been described, including in the papers by Srinivasin, Nelson, and Yeoman that have been cited, but moreover have been quantitated as influencing ~25 - 31% of metabolomic variation, see Borgogna et al. 2020 BJOG 127:182 (<https://doi.org/10.1111/1471-0528.15981>) and Borgogna et al. 2018. Sci Rep. 8: 852 (<https://doi.org/10.1038/s41598-017-14943-3>) greater introduction around this fact and discussion, including comparisons to your findings herein are warranted.

Further, Ilhan et al. 2019 EBioMedicine 44: 675 (<https://doi.org/10.1016/j.ebiom.2019.04.028>) look at the relationship between the cervicovaginal metabolome and immune response and similarly should be discussed and compared.

line 108 - 109: The influence of the differing primer sets deserves greater examination/discussion - is B. breve dominant in this CST? and is this artifactual or real?

line 257: '>M<asses'?

Point-by-point reply to reviewer comments

Reviewer #1 (Remarks to the Author):

- *This study investigates how the vaginal microbiome contributes to risk of preterm birth, the primary cause of death in children under 5 years of age, using metabolic profiling by DESI-MS for characterization of the cervicovaginal metabolome. The authors describe how the metabolome signatures can be used to predict the composition of the vaginal microbiome and inflammatory status. The primary findings focus on using on-swab metabolic DESI-MS profiling for rapid preterm birth risk stratification, a very important area of study.*

The study design is generally good with two separate patient cohorts seen as a major positive. Although it was surprising that the identified metabolites were not further validated/quantified using a separate LC/QqQ based method, essentially the gold stand for such analyses. Yet the use of LC/MS QTOF data is compelling.

While I usually approach these desorption/ionization studies with a fair degree of skepticism, I am impressed with the level of rigor that went into this effort. While I would have preferred to see the QqQ analyses as a level of validation, nonetheless this is a solid paper. Although how the immunological data was used is also a bit confounding and how all the data from DESI, LC/MS, lipid profiling, combined with the immunological data was not clear.

The figures are well done and the data is compelling yet their interconnectivity is a bit confusing.

We thank the reviewer very much for acknowledging the effort and level of experimental rigor we undertook as part of our study designed, which included where possible, validation of the major findings in two independent patient cohorts.

LC-MS assays described in our study were performed using qTOF instrumentation as a parallel untargeted metabolomics investigation to compare with the DESI-MS prediction capacity. As we did not have a list of predefined metabolite biomarkers to measure at the time, we could not develop a targeted LC-QqQ method. In addition, the number of biomarkers that we ultimately discovered and replicated is large (ie. >100 for microbial composition prediction) thus it was not feasible within the scope of this project to develop a single, multiplexed LC-QqQ method.

Following the comments from reviewer, we have amended the manuscript to improved the description of how our analyses were performed and how the data was interlinked within the context of microbial-host interactions during pregnancy and preterm-birth (see throughout manuscript).

- *Page 3 desorb should be desorb.*

This has now been corrected.

Reviewer #2 (Remarks to the Author).

Major Points:

1. *It is unclear why DESI-MS was not used directly to predict pre-term birth within the cohorts assessed. Instead, DESI-MS measurements were used to predict bacterial composition/cytokine production, which was then discussed as an important factor in pre-*

term birth. DESI-MS does seem like a useful tool for gaining biological insight into the relationships between the microbiome and the host immune response in pregnancy, though this does not seem to be the focus of the manuscript. What is the motivation of using DESI-MS to predict immune marker concentration, and then using the predicted immune marker concentration to associate with pre-term birth classification?

We thank the reviewer for highlighting this oversight and agree that the transition from describing the predictive capacity of DESI-MS through to highlighting its potential value in the context of different clinical scenarios associated with preterm birth was not clearly presented. Further, we agree that a more direct analysis of the ability of DESI-MS to predict preterm birth within these patient cohorts is justified.

The revised manuscript now contains a series of analyses where random forest modelling was used to predict preterm birth using the information in the cervicovaginal DESI-MS metabolic profiles as well as metataxonomics and inflammatory marker data. This has been tested at 3 gestational timepoints in pregnancy. Overall, the predictive capacity of these models was poor (Supplementary Data Table 6). These results suggest that neither the cervicovaginal fluid metabolome as measured with DESI-MS, bacterial metataxonomics profile nor the inflammatory markers measured in our study contain sufficient information to robustly predict preterm birth. Given the fact that preterm birth is a multi-aetiological disease state that can be caused by many different factors (including non-microbial), this finding is somewhat expected. In light of these findings, we then focused subsequent analyses on women who receive cervical cerclage with braided suture material, who we have previously shown are at increased risk of preterm birth that involves a phenotype characterised by vaginal dysbiosis and local immune activation. In this patient cohort, our results highlight the potential utility of DESI-MS swab analysis for monitoring microbiota and inflammatory responses in a way that could be transformative for directing clinical decision making and ultimately improving outcomes for these women and their babies.

These new findings have now been reported in the manuscript (see Results, page 8, paragraph 1; Supplementary Table 6) and their significance are described in the discussion (see Discussion, page 10, paragraph 2).

- 2. If purpose of this manuscript is to introduce a new technology for diagnostic, it could be helpful to include more text in the introduction and discussion sections regarding why this methodology is superior to previous techniques (if any) to predict PTB or whether the goal is a new method to understand multi-omics relationships between metabolome, vaginal microbiome composition and inflammation. If the latter, how is it better than traditional techniques to quantify metabolome (RP-LC-MS, HILIC-LC-MS)? Additionally, if it is being proposed as a new diagnostic tool, how do prediction capabilities feasibility compare to the use of vaginal microbiota composition or a panel of immune biomarkers (say IL-1beta and MBL) to predict risk of PTB?*

We thank the reviewer for this suggestion. We have now included additional text in the introduction and discussion regarding the need for a new point-of-care diagnostic that reports relationships between metabolome, vaginal microbiome composition and inflammation (see Introduction, page 3, paragraph 1; Introduction, page 4, paragraph 1). We also better highlight the advantages of DESI-MS over traditional metabolomics platforms in addition to drawing attention to the benefit of acquiring simultaneous information on both vaginal microbiota composition and immune status without the

need of performing additional expensive and comparatively laborious assays (see Introduction, page 4, paragraph 2; Discussion, page 10, paragraphs 1 and 2) .

- 3. Were repeated measures considered in the random forest models? Pregnancy term influences vaginal microbiota composition and metabolite changes may also be dependent on time point in pregnancy.*

The random forest models used for prediction of vaginal microbiota composition used all available samples from each individual. Gestational age was not originally included in the random forest models that were reported, but was included in the linear mixed effect models. We have now repeated the random forest models using gestational age as an extra predictor. This had no effect on the ability of the models to predict microbial composition (see Figure 1 below). These models and their results are now described in the manuscript (see Methods, “Prediction of CST, preterm birth and immune markers from the metabolomic data”, page 21) and are reported in full as part of the GitHub code repository for the data analysis <https://www.github.com/gscorreia89/desi-ms-cst-typing/>.

The original decision to not include gestational age in the random forest models was supported by linear mixed effect analyses that showed almost no overlap between the metabolomic signatures used for microbiota compositional prediction and gestational age. This information is also now presented in the methods as well as Supplementary Figure 2. We further support this by undertaking an additional detailed assessment of metabolome variance explained by CST, Gestational Age, Ethnicity, BMI and maternal age (Supplementary Data Figure 3 and Supplementary Table 3).

VMET

VMET2

Figure 1. Comparison of cross-validated classification metrics and their distribution in models trained to predict LDOM/LDEPL, with (withGA) or without (w/ GA) gestational age as a predictor variable. A) VMET DESI-MS (-); B) VMET DESI-MS (+); C) VMET2 DESI-MS (-); D) VMET2 DESI-MS (+). The mean estimates (black dot) and their distributions (boxplot), are similar for both ionisation modes in VMET and VMET2, showing that incorporating gestational age as predictor in the random forest models does not improve the classification of LDOM/LDEPL.

4. *For the selection of a signature to predict LDOM/LDEPL, how was the signature validated? Was all the data from both cohorts used in the linear mixed effect model that suggested the 113 metabolite features? Or was the signature validated in only a specific cohort?*

The linear mixed model data analysis used for identifying the LDOM/LDEPL signatures was independently performed in both the VMET and VMET2 cohort. The linear mixed effect modelling and FDR correction for the VMET cohort used only VMET data, and vice-versa. After obtaining the

Benjamini-Hochberg FDR controlled signatures for each study, we then matched the m/z values from each final list of significant features, with a maximum m/z error tolerance of 5 ppm. A metabolic association was considered to be replicated if it was present in the list of final FDR adjusted hits of both cohorts, for the same contrast. In addition to updating the methods (see Methods, “Statistical analysis of metabolomic profiling data”, pages 20-21), we have provided additional description about the validation approach in the manuscript (see Results, “Prediction of vaginal microbiota composition by direct on-swab DESI-MS, page 5) as well as in the legend of Figure 1B.

5. *For the in vitro measurements, why were multiple strains of Lactobacillus spp. evaluated compared to single strains of bacteria associated with CST IV? This may be important since G. vaginalis is a highly variable species and L. gasseri and L. jensenii were not observed as frequently in the samples. L. iners strains can also be quite variable and were the next most common Lactobacillus sp. after L. crispatus in the samples evaluated.*

The strains used for the *in vitro* measurements were limited to those available in house. We agree with the reviewer that this represents a restricted list of bacterial isolates that are not necessarily representative of the genetic diversity of vaginal commensal and pathogenic strains. This has now been clearly acknowledged in the text (see Discussion, page 10, paragraph 2)

6. *The discussion (p. 9) states that this approach could be extended for bedside treatments designed to optimize VMC. It could be interesting to assess the use of DESI-MS compared to standard practices to diagnose VMC (typically for BV) such as Nugent Score, Amsel Score, or 16S qPCR in regards to accuracy, cost, speed of result and ease of use by clinician (in clinic) or by patient (home self-swab). More discussion in this direction would be valuable.*

We thank the reviewer for this suggestion and have included additional discussion on how direct swab profiling by DESI-MS compared to other standard practices used for diagnosing vaginal bacterial communities (see Discussion, page 10, paragraphs 1 and 2).

Minor Points and Suggestions:

1. *Table 1: define “i” and “s” PTB*

iPTB and sPTB have now been defined (see Table 1).

2. *Figure 1c: add statistical significance bars or p-values to plots.*

We have now added p-values (corrected) to the plots as requested.

3. *Figure 4b: change DESI-MS m/z features to names of compounds.*

This DESI-MS m/z features have been replaced with compound names as requested.

4. *Fig S2 and Fig S3: text on the plots is illegible.*

The resolution and size of text on these figures has been modified to ensure that they are now clearly legible.

5. *Github link is not public (<https://www.github.com/qscorreia89/desi-ms-cst-typing/>).*

The Github link is now public.

6. It is unclear whether results support that DESI-MS allows for species level classification since the most resolution demonstrated was to predict CSTs, and thus only applies to *Lactobacillus* spp.

At this stage our data supports CST level classification. We expect that as larger DESI-MS and metataxonomic datasets are acquired and better representation of more diverse compositions and rarer taxa are obtained, species level classification may be achievable. This is now noted in the discussion (see page 12).

Reviewer #3 (Remarks to the Author):

The paper describes the use of a Direct on-swab metabolic profiling approach as a potential 'bedside' point of care testing technique for examining cervicovaginal microbiota and host immune activity related to pregnancy and preterm birth risk. It is an interesting and, potentially valuable addition to the literature. The manuscript is mostly well written and I only have several minor comments that should be addressed.

We thank the reviewer for the supportive comments.

Line 50: "...increased >microbial< diversity..."

We have now corrected this in the manuscript.

Lines 51-52 Not sure this is worded how you intend it to ("...and the effective treatment of STI...") - appears to suggest lactobacillus depletion/BV is associated with more effective treatment of STI.

Thank you for highlighting this. This has now been corrected.

Lines 76 - 79: The influence of bacterial composition on the metabolome has been described, including in the papers by Srinivasin, Nelson, and Yeoman that have been cited, but moreover have been quantitated as influencing ~25 - 31% of metabolomic variation, see Borgogna et al. 2020 BJOG 127:182 (<https://doi.org/10.1111/1471-0528.15981>) and Borgogna et al. 2018. Sci Rep. 8: 852 (<https://doi.org/10.1038/s41598-017-14943-3>) greater introduction around this fact and discussion, including comparisons to your findings herein are warranted.

We thank the reviewer for this suggestion. We have now undertaken additional analyses of the metabolome variance partitioning. Linear mixed effect model-based (semi-partial and conditional r^2 values) and PERMANOVA analyses were performed to obtain a breakdown of variance per covariate (Supplementary Fig. 3 and Supplementary Table 3). These results show agreement with previous findings in the literature, and as suggested by the reviewer, our variance estimates are now discussed in the context of the important previous reports by Borgogna and colleagues within the main text of the manuscript (see Discussion, page 9, paragraph 3).

Further, Ilhan et al. 2019 EBioMedicine 44: 675 (<https://doi.org/10.1016/j.ebiom.2019.04.028>) look at the relationship between the cervicovaginal metabolome and immune response and similarly should be discussed and compared.

We thank the reviewer for bringing this interesting reference to our attention. We have included this reference and a comparison with our results in the discussion (see Discussion, page 9, paragraph 2.)

line 108 - 109: The influence of the differing primer sets deserves greater examination/discussion - is B. breve dominant in this CST? and is this artifactual or real?

The detection of *B. breve* dominated CSTs in the VMET2 cohort is thought to be real. The mixed formulation of the 27F forward primer set used in the VMET2 patient cohort has been shown to maintain the rRNA gene ratio of key vaginal species including *Lactobacillus* spp. to *Gardnerella* spp. as well as improve detection of Bifidobacterial species, which are otherwise not detected by the “universal” 27f primer often used in metataxonomics studies (Frank et al., Applied and Environmental Microbiology. 2008;74(8):2461-70; Walker et al., Microbiome volume 3, Article number: 26 (2015). These results are also consistent with other studies of the vaginal microbiome (e.g. France et al., Microbiome volume 8, Article number: 166 (2020); Lee et al., Front. Public Health 8:507024 (2020)). We have now included this important additional discussion around the different primer sets used in the study cohorts (see Discussion, page 12).

line 257: '>M<asses'?

This has now been corrected.

REVIEWERS' COMMENTS

Reviewer #2 (Remarks to the Author):

I appreciate the authors thoughtful response to the reviews, the rigor of the validation cohort, and the changes to the manuscript. It does seem that some of the results/abstract are still overstated in terms of the ability to predict pre-term birth risk, but this could be modified. This reviewer also recommends checking the Github link, which doesn't seem to be working.

Reviewer #3 (Remarks to the Author):

The reviewers have sufficiently addressed all of my concerns and I feel the manuscript is suitable for publication.

Point-by-point reply to reviewer comments

Reviewer #2 (Remarks to the Author):

I appreciate the authors thoughtful response to the reviews, the rigor of the validation cohort, and the changes to the manuscript. It does seem that some of the results/abstract are still overstated in terms of the ability to predict pre-term birth risk, but this could be modified.

We thank the reviewer for the kind remarks and for helping to improve the manuscript during the review process. During the previous round of reviews, we substantially altered the language used throughout the paper to ensure we did not overstate our findings. This included performing additional analyses designed to directly test the ability of DESI-MS and other metabolic profiling approaches to predict preterm birth. In the paper we state, “We next tested if DESI-MS metabolic, metataxonomics and inflammatory marker profiles obtained at three different stages of gestation, could predict subsequent preterm birth. Overall, the predictive capacity of these models was poor (Supplementary Table 6).” We do not believe that this could be interpreted by readers as overstating the ability to predict preterm birth. Our claims that direct on-swab metabolic profiling “can be used to robustly predict simultaneously both the composition of the vaginal microbiome and host inflammatory status” are supported by the data we have presented. Given the well-described and important role for the vaginal microbiome and inflammation in mediating specific preterm birth risk phenotypes, we feel that our method does indeed provide “an innovative approach for preterm birth risk stratification through rapid assessment of vaginal microbiota-host dynamics”, as we conclude in the abstract.

This reviewer also recommends checking the Github link, which doesn't seem to be working.

Thank you for bringing this to our attention. We have now corrected the typo in the Github link and have confirmed that it is active and publicly available.

Reviewer #3 (Remarks to the Author):

The reviewers have sufficiently addressed all of my concerns and I feel the manuscript is suitable for publication.

We thank the reviewer for the kind remarks and appreciate the time and effort made to improve our manuscript.